# Pervasive cooperative mutational effects on multiple catalytic enzyme traits emerge via long-range conformational dynamics

Carlos G. Acevedo-Rocha[1,11], Aitao Li [2,11], Lorenzo D'Amore [3,11], Sabrina Hoebenreich [4], Joaquin Sanchis [5], Paul Lubrano[1,10], Matteo P. Ferla[6], Marc Garcia-Borràs [3], Sílvia Osuna [3,7✉] & Manfred T. Reetz[4,8,9✉]

Multidimensional fitness landscapes provide insights into the molecular basis of laboratory and natural evolution. To date, such efforts usually focus on limited protein families and a single enzyme trait, with little concern about the relationship between protein epistasis and conformational dynamics. Here, we report a multiparametric fitness landscape for a cytochrome P450 monooxygenase that was engineered for the regio- and stereoselective hydroxylation of a steroid. We develop a computational program to automatically quantify non-additive effects among all possible mutational pathways, finding pervasive cooperative signs and magnitude epistasis on multiple catalytic traits. By using quantum mechanics and molecular dynamics simulations, we show that these effects are modulated by long-range interactions in loops, helices and β-strands that gate the substrate access channel allowing for optimal catalysis. Our work highlights the importance of conformational dynamics on epistasis in an enzyme involved in secondary metabolism and offers insights for engineering P450s.

[1] Biosyntia ApS, Copenhagen, Denmark. [2] State Key Laboratory of Biocatalysis and Enzyme Engineering, Hubei Collaborative Innovation Center for Green Transformation of Bio-Resources, Hubei Key Laboratory of Industrial Biotechnology, School of Life Sciences, Hubei University, Wuhan, P. R. China. [3] Institut de Química Computacional i Catàlisi and Departament de Química, Universitat de Girona, Girona, Spain. [4] Department of Chemistry, Philipps-University Marburg, Marburg, Germany. [5] Monash Institute of Pharmaceutical Sciences, Monash University, Parkville, VIC, Australia. [6] Wellcome Centre for Human Genetics, University of Oxford, Oxford, UK. [7] ICREA, Barcelona, Spain. [8] Department of Biocatalysis, Max-Planck-Institut für Kohlenforschung, Mülheim an der Ruhr, Germany. [9] Tianjin Institute of Industrial Biotechnology, Chinese Academy of Sciences, Tianjin, P. R. China. [10]Present address: Bacterial Metabolomics Group, Eberhard Karls Universität Tübingen, Tübingen, Germany. [11]These authors contributed equally: Carlos G. Acevedo-Rocha, Aitao Li, Lorenzo D'Amore. ✉email: silvia.osuna@udg.edu; reetz@mpi-muelheim.mpg.de

Directed evolution constitutes a powerful tool for optimizing protein properties, including activity, substrate scope, selectivity, stability, allostery or binding affinity. By applying iterative rounds of gene mutagenesis, expression and screening (or selection), proteins have been engineered for developing more efficient industrial biocatalytic processes[1–4]. Directed evolution has also provided important insights into the relationship between protein sequence and function[4–6], yet understanding the intricacies of non-additive epistatic effects remains a challenge[7]. Epistasis means that the phenotypic consequences of a mutation depend on the genetic background[8–11]. Epistatic effects can be negative (antagonistic/deleterious) or positive (synergistic/cooperative) if the respective predictive value is smaller or greater in sign/magnitude than the expected value under additivity. Sign epistasis (SE) occurs when a mutation has a deleterious or beneficial effect alone but an opposite effect when combined with other(s) mutation(s), whereas in magnitude epistasis (ME) a mutation has a deleterious or beneficial effect in isolation and in combination with other mutation(s). Based on studies of natural and laboratory protein evolution, negative[10] or positive[11] epistasis is more widespread than originally thought[7]. Importantly, positive epistasis increases the evolution of new protein functions because it allows access to mutational pathways that avoid deleterious downfalls. On the other hand, negative epistasis has been associated with a higher tolerance for mutations, which is important because this mutational robustness enables protein stability and evolution[12]. For fundamental and practical reasons, it is thus important to determine the existence, type and molecular basis of epistasis in protein evolution. Epistatic effects can arise between residues that are located closely or away from each other via long-range indirect interactions, both mechanisms involving sometimes direct or indirect substrate binding[11]. These global epistatic effects may be mediated by changes in the protein conformational dynamics.

Proteins have the inherent ability to adopt a variety of thermally accessible conformational states, which play a key role in protein evolvability and activity[13,14]. Along the catalytic cycle, enzymes can adopt multiple conformations important for substrate binding or product release[15,16], and conformational change can be rate-limiting in some cases[17,18]. Much debated is the existence of a link between active site dynamics and the chemical step[19,20]. Some studies have suggested that mutations remote from the enzyme active site may directly impact the energetically accessible conformational states, thereby influencing catalysis[21–23]. This has been shown by means of crystal structures and nuclear magnetic resonance (NMR) spectra of mutants along evolutionary pathways[21,24,25] together with computational assistance[26–30]. Molecular dynamics (MD) simulations, which are highly complementary to NMR analyses[31], allow the partial reconstruction of the enzyme conformational landscape, and how this is altered by mutations introduced by laboratory evolution[29,30,32]. Tuning the enzyme conformational dynamics can play an important role in the emergence of novel activities[22,25,30,32,33].

The connection between conformational dynamics and epistasis has been studied in proline isomerase (cyclophilin A)[24], phosphotriesterase[34] and β-lactamases[35–37]. For example, negative SE between two distal mutations limited dynamics of active site loops mediating substrate accessibility in a β-lactamase[35]. These studies provide fascinating insights, but they are limited to a single protein trait (usually activity) as a measure of fitness. This term originally refers to the reproductive success of organisms, but it can be applied to protein activity, selectivity or stability[34–37]. This contrasts with directed evolution where often two or more traits (e.g. activity and selectivity or stability) are sought for practical purposes[4,38]. Therefore, connecting epistasis to conformational dynamics increases our understanding of proteins. In turn, analysing non-additive epistatic effects can be expected to benefit in silico-directed evolution[39].

In the present work, we used a combination of enzyme kinetics and computational approaches to investigate epistatic effects and conformational dynamics in the stepwise evolution of a cytochrome P450 monooxygenase (CYP) engineered for the highly active, regioselective and stereoselective oxidative hydroxylation of a steroid as a non-natural substrate[40].

To determine epistatic effects effectively, we developed a Python-based script and freely accessible web-app (https://epistasis.mutanalyst.com/), which can be used for any enzyme and catalytic trait (or for any protein and parameter). Unexpectedly, we found pervasive positive epistatic effects on multiple catalytic traits, with selectivity and activity being generally characterized by SE and ME. We found that the analysis of the link between protein epistasis and conformational dynamics reveals the increasing optimization of activity and selectivity along all evolutionary trajectories through fine tuning of loops, helices and β-strands that gate active site entrance and modulate the active site by long-range networks of interactions. Our study offers guiding principles for the simultaneous engineering of both activity and selectivity in a model CYP member.

## Results

**Multiple parameters define the biocatalytic landscape in P450$_{BM3}$.** The CYP super protein family has >300,000 members that are involved, among others, in the biosynthesis of steroids, fatty acids and natural products as well as in the degradation of drugs in humans and of xenobiotics in the environment.[41,42] Thus this is an important enzyme class with relevant applications in biocatalysis, biomedicine, pharmacology, toxicology and biotechnology[42–44]. Previously, we achieved the stereoselective and regioselective hydroxylation of testosterone (**1**) by evolving the self-sufficient *Bacillus megaterium* cytochrome P450$_{BM3}$ monooxygenase[40]. P450$_{BM3}$ is one of the most active and versatile CYPs that oxidizes long fatty acids as the natural substrates, and there are various three-dimensional structures of the haem domain alone without the reductase domain[45]. However, wild type does not accept steroids, which is the reason why we chose mutant F87A as the starting enzyme. While F87A accepts **1**, it provides in a whole-cell system only ~20% conversion with formation of a 1:1 mixture of 2β-hydroxytestosterone (**2**) and 15β-hydroxytestosterone (**3**)[40]. Combinatorial saturation mutagenesis at the randomization site R47/T49/Y51 allowed the evolution of mutant R47I/T49I/Y51I/F87A (III) displaying 94% 2β-selectivity and 67% conversion of **1** (1 mM) in 24 h whole-cell reactions[40] (Fig. 1a). The mechanism is known to involve a radical process in which the catalytically active haem-Fe=O (Cpd I) abstracts an H atom from aliphatic C-H followed by a fast C-O bond formation, which requires a precise substrate positioning, as in other cases[46] (Supplementary Fig. 1). The three mutated residues are located next to each other (distances of C$_\alpha$ is ~6 Å) lining the large binding pocket but relatively far away (~15–20 Å) from haem-Fe=O, assuming the absence of dynamic effects (Fig. 1b). Complete deconvolution of variant III starting from parental F87A entails $3! = 6$ theoretical pathways, which we constructed by generating the respective 6 intermediate mutants (Fig. 1c/d). The key question is how these residues determine selectivity and activity and whether they interact epistatically.

All intermediate mutants were generated, overexpressed in *Escherichia coli* BL21-Gold(DE3) and purified (Supplementary Fig. 2). Parent F87A (---) and variant III were also included, resulting in a total of 8 enzymes. Using defined substrate and NADPH concentrations, multiple parameters

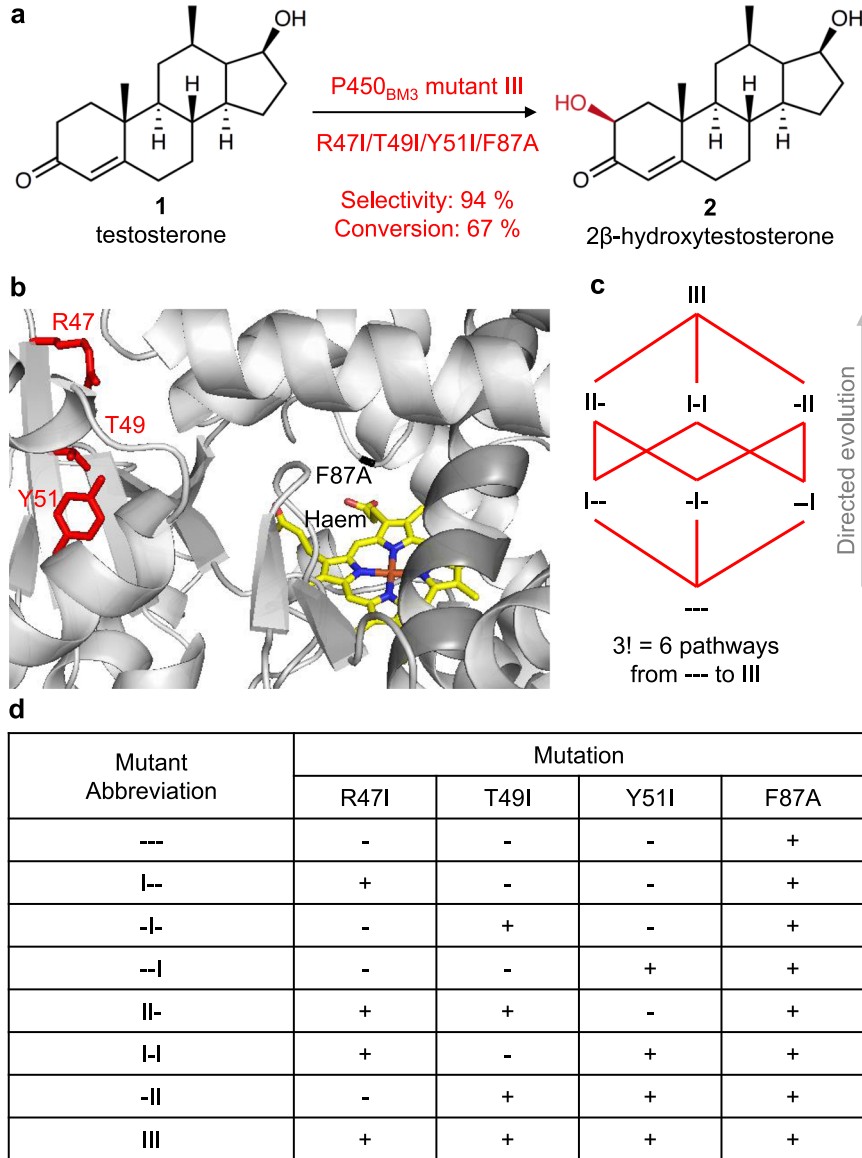

**Fig. 1 Model system based on P450$_{BM3}$ as biocatalyst for the selective oxidation of a steroidal substrate. a** Testosterone (**1**) is selectively hydroxylated at position 2β (**2**) by mutant III (R47I/T49I/Y51I/F87A). **b** Active site of parent enzyme showing F87A mutation (black) above the haem (yellow) as well as WT resides R47, T49 and Y51 (red). The distances between the α-C atoms of the following pairs of residues are (Å): R47-T49 (7.0), R47-Y51 (13.4), T49-Y51 (7.0). Image and atom distances calculations obtained with PyMol Molecular Graphics System, V. 1.5.0.4 (Schrödinger, LLC). An interactive figure of parental variant --- docked with **1** was created with Michelanglo[68] (https://michelanglo.sgc.ox.ac.uk/r/p450) highlighting the mutated residues and secondary structures discussed in this work. **c** The 6 possible evolutionary trajectories between parental mutant F87A (---) and "triple" mutant III involve three "single" mutants I-- (R47I/F87A), -I- (T49I/F87A) and --I (Y51I/F87A) as well as three "double" mutants II- (R47I/T49I/F87A), I-I (R47IY51I/F87A) and -II (T49I/Y51I/F87A). **d** Mutant abbreviations. The signs − and + indicate that the respective mutation is absent and present, respectively.

were determined (Supplementary Note 1 and Supplementary Table 1).

The most 2β-selective variants (~67–91%) contain mutation Y51I (--I, I-I, -II and III), while the remaining ones proved to be 15β-selective, with substrate conversion being highest (35%) in mutants III and -II, and poor (~6–10%) in the remaining ones (Fig. 2a). NADPH leak rate without substrate is higher than NADPH consumption rate (NCR) in all mutants except -II and III, which display a respective <2- and <3-fold increased NCR (Fig. 2b). Mutants -II and III also showed a respective ~5- and ~10-fold improvement in product formation rates (PFRs) compared to the remaining variants (Fig. 2c), suggesting that variants -II and III have a good coupling efficiency (CE). CE describes how well the reductase domain delivers electrons from

NADPH via the flavin cofactors to the substrate in the haem domain. A low CE value indicates futile NADPH usage, resulting in the formation of reactive species during the catalytic cycle (Supplementary Fig. 1) that can inactivate the enzyme[47]. Low CE values of 15–30% were found for all mutants, except for -II and III that display higher values of 37% (Fig. 2d). The total turnover number (TTN) is highest in mutants -II and III (Fig. 2e), whereas the total turnover frequency (TTF), PFR and NCR are highest in III (Fig. 2f).

**Distal mutations enable conformational changes at the active site required for regioselectivity.** To gain insights about the origin of selectivity and activity, we performed computational

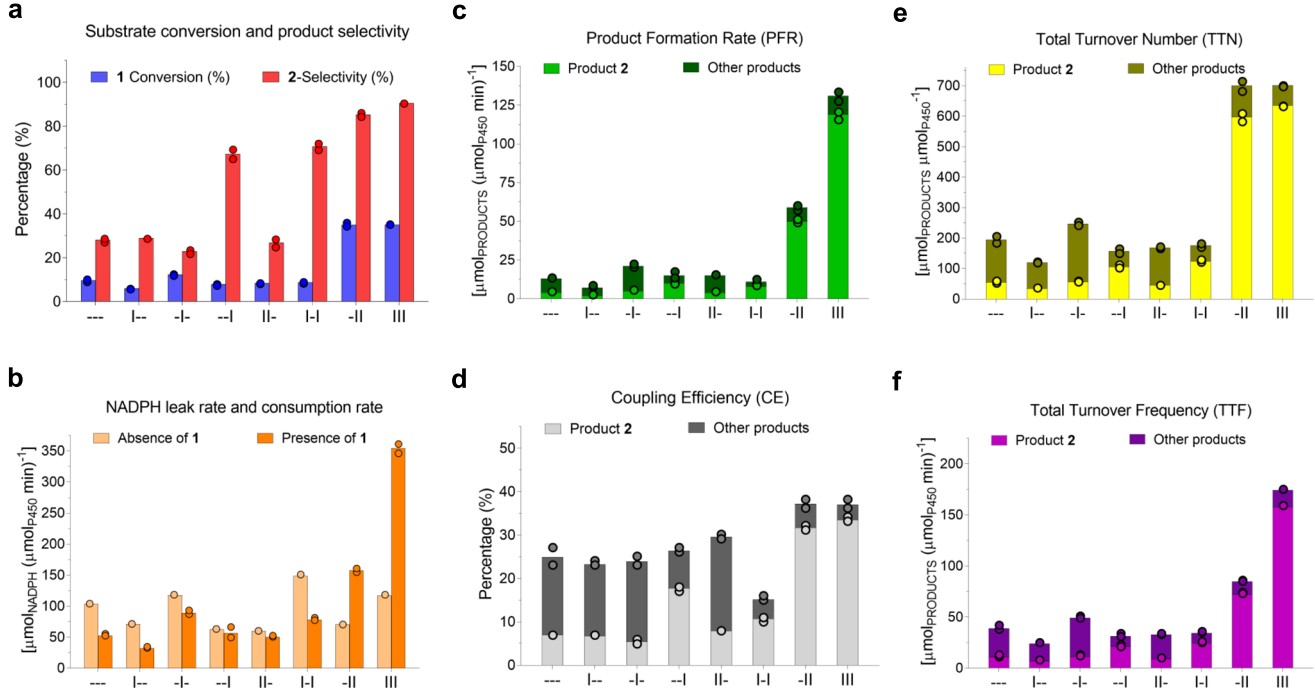

**Fig. 2 Multiple enzymatic parameters of deconvolution mutants. a** Selectivity and conversion data are obtained from HPLC data and shown in percentage. **b** NADPH leak and consumption rate were measured in the absence and presence of testosterone (**1**) substrate, respectively. **c** Product formation rate (PFR) is calculated by multiplying the NADPH consumption rate by coupling efficiency. **d** Coupling efficiency (CE) is the ratio between NADPH consumption and production formation, and it is reported in percentage. **e** TTN describes the total moles of products per moles of enzyme after NADPH depletion. **f** TTF normalizes TTN by time after NADPH depletion. See Fig. 1d for mutant abbreviations. Other products mainly include the 15β-alcohol and other regioisomers. The data represent the average of two independent experiments ($n = 2$). Source data are provided with this paper.

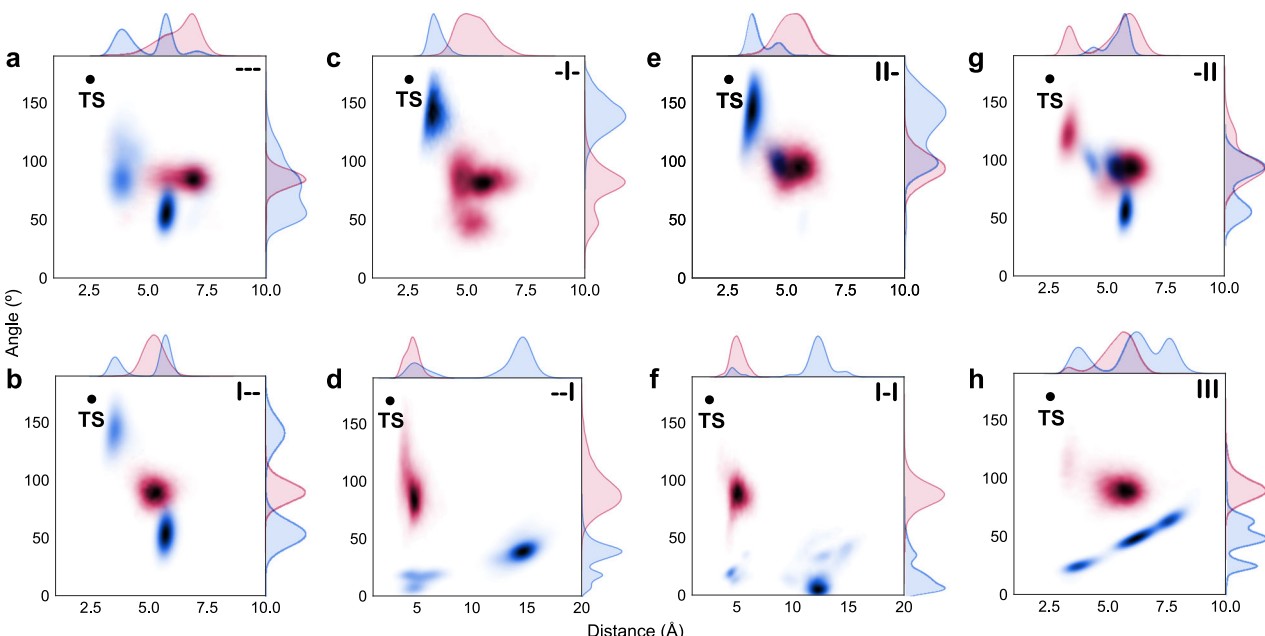

**Fig. 3 Conformational population analysis of key geometric parameters for hydroxylation.** Distances determined between the oxygen atom of haem-Fe=O and the C-atom (C2 or C15) of **1** (x-axis) and angles formed by O(Fe = O) – (**1**)-H(C2/15) – (**1**)-C(2/15) (y-axis) from the first replica of the MD dataset of parent mutant (**a**), "single" mutants (**b–d**), "double" mutants (**e–g**) and "triple" mutant III (**h**) (see Supplementary Figs. 6 and 7 for additional replicas). Geometric parameters measured for C-2 and C-15 are shown in red and blue, respectively. The ideal distance and angle for the transition state TS (black dot) corresponds to the Density Functional Theory (DFT) optimized geometry for the C–H abstraction by haem-Fe=O using a truncated computational model (Supplementary Note 2). See Fig. 1d for mutant abbreviations.

studies on all mutants. Given the identification of comparable reaction barriers for hydrogen atom abstraction from C2 and C15 by using Density Functional Theory (DFT) calculations on truncated models (difference of <1.0 kcal/mol, see Supplementary Note 2, Supplementary Fig. 3 and Supplementary Table 2), we carried out MD simulations. For each mutant, we started from pose 15 and from pose 2 (i.e. positioning C15 or C2 closer to the Fe=O, respectively) to analyse whether the binding pose of 1 in the active site determines the experimentally observed selectivity (Fig. 3 and Supplementary Note 3). Starting from parent ---, pose 2 (presenting C2 close to the catalytic Cpd I) and pose 15 (C15 close to Cpd I) generated from manual dockings are possible (Supplementary Figs. 4 and 5). Further analysis of these binding poses along MD simulations in --- indicate that substrate 1 in pose 15 explores near attack conformations (NACs)[48] closer to the quantum mechanics-predicted ideal transition state geometry for H abstraction than in pose 2 (Fig. 3a), thus making pose 15 more productive towards 15β-hydroxylation. Introducing mutations R47I and/or T49I does not have any effect on selectivity (Fig. 3b, c, e), i.e. the selectivity is retained due to the catalytically competent conformation inherent in pose 15 along MD simulations (pose 2 adopts a reduced number of catalytically competent conformations). However, the picture completely changes when mutation Y51I is introduced: the substrate bound in pose 15 becomes unstable and leaves the active site in 1 out of 3 replicas (ca. >15 Å C2··O distances explored, Fig. 3d), whereas pose 2 is highly stabilized and explores short C2··O distances for the incipient C-H eventually leading to 2β-hydroxytestosterone in 2 out of 3 replicas (Fig. 3d and Supplementary Figs. 6 and 7). As experimentally determined, 2β-selectivity is retained in variants I-I, -II and III that contain mutation Y51I (Fig. 3f–h). This is even more dramatic in variant III, in which pose 15 is highly unstable and 1 rapidly rotates to position C2 close to the catalytic Cpd I for 2β-hydroxylation (Supplementary Movie 1). Instead, pose 2 in variant III is stable and adopts near attack conformations in all MD replicas (Fig. 3d and Supplementary Figs. 6 and 7).

Notwithstanding, mutant -II and III only differ for the R47I mutation in the latter case, yet the re-orientation of the substrate from pose 15 to pose 2 is observed only during the MD simulation of mutant III. To further investigate the specific effect of R47I mutation on substrate rotation inside the haem pocket, we performed a Principal Component Analysis (PCA) on the substrate-bound MD trajectories of mutant -II and III, finding that pc2 indeed describes an increased flexibility of residues A87, T260, G265 and T327 in mutant III, as compared to -II (Supplementary Fig. 8). Thus R47I may modulate via long-range conformational dynamic effect the flexibility of such residues, which have been shown to be instrumental to promote substrate re-orientation in mutant III (Supplementary Movie 2). Moreover, mutant III presents a substantially wider active site pocket as compared to the other variants: the active site volume in the --- variant is 89 Å$^3$, which is expanded to 235 Å$^3$ in III (Supplementary Fig. 9). We hypothesized that, in all variants, except III, selectivity must be determined by the orientation adopted by the substrate while accessing the haem cavity. Recently, Mondal et al. characterized the substrate recognition and binding pathway in related P450cam using MD simulations, showing the formation of a single key channel in which the substrate needs to reside in a long-lived intermediate state before reaching the catalytic iron-oxo species[49].

To reconstruct the substrate-binding process in P450$_{BM3}$, we placed substrate 1 in the bulk solvent and started unbiased MD simulations followed by accelerated molecular dynamics (aMD) simulations (Supplementary Note 4). Among independent MD and aMD trajectories for all variants (250 ns MD + 750 ns aMD), only a single trajectory by mutant I-- was observed to be productive where

the substrate reached the haem active site (Fig. 4b). We observed a two-step binding mechanism in this trajectory: first, the carbonyl moiety of 1 enters channel 2a (Fig. 4a) and stays above the β1-2 strand, where residues R47I, T49I and Y51I are located, forming a long-lived substrate–enzyme bound intermediate. There, substrate 1 can reorient, although its access to the active site is restricted by the β4 sheet that acts as a gate. Second, a network of coupled conformational changes occur simultaneously: G helix adopts a bend conformation, which impacts F helix, F–G loop and β1 sheet conformation, and in turn retreats β4 sheet, allowing 1 progression towards the catalytic centre (Supplementary Video 2). This two-step mechanism is similar to what Mondal et al. observed in P450cam[49].

Our aMD simulations indicate that the orientation of the substrate when accessing the catalytic site during the second step of the binding pathway dictates selectivity. Once inside the haem pocket, substrate rotation was not observed in mutant I--, thus predicting that the orientation assumed by the substrate when entering the haem site ultimately governs selectivity. In fact, the previous substrate-bound MD simulations suggest that only variant III has a sufficiently wide active site pocket for allowing substrate rotation. These findings indicate that, in all variants, except III, selectivity is determined by the orientation adopted by the substrate while entering the haem pocket. In the productive trajectory corresponding to mutant I--, 1 accesses the haem with the correct orientation for 15β-hydroxylation (Fig. 4b and Supplementary Fig. 10). In this case, residue Y51 establishes a hydrogen bond with the carbonyl group of 1 (Supplementary Fig. 11), constraining the substrate in such way that it can only progress into the active site pocket pointing its C15 ahead towards haem-Fe=O[50]. Thus Y51 is instrumental in promoting the observed C15-selectivity in --- and in I--, -I- and II- variants. It should be noted that, in previous studies, R47 and especially Y51 were found to interact with the terminus end of long-chain fatty acids while bound at the P450$_{BM3}$ active site[51]. Such direct interaction with testosterone and Y51 is only possible at the pre-binding pocket, which is lost after the retreat of β4 sheet, allowing substrate access to the haem pocket. Additionally, the higher C2-selectivity observed in variant III occurs due to the flipping and motion of the β4 sheet, destabilizing pose 15 while favouring pose 2 (Fig. 4c).

Interestingly, the analysis of the most relevant conformational changes in each independent variant through PCA predicts that the most active mutant III shows the highest flexibility of the β4 sheet (Fig. 4d). This higher flexibility related to activity has, however, no impact on the B' helix and B-B' loop conformational dynamics (Supplementary Figs. 12 and 13). These flexible regions, responsible of controlling substrate binding as described above, are likely to influence activity, as mutant III shows the highest TTF, NCR and PFR numbers. These findings suggest that favouring a more efficient substrate binding in a catalytically competent pose increases enzyme TTF, while NADPH leak is reduced due to a more efficient interaction between the substrate and the catalytically active Fe=O species once generated.

**Pervasive epistatic effects on multiple parameters are cooperative.** According to Tokuriki[11] and Bendixsen et al.[52], non-additive mutational effects can occur in different forms (Fig. 5), which can be calculated with additivity equations (Supplementary Note 5). Aiming at exploring the existence of epistatic effects in an effective manner, we developed and applied a Python-based computational program to automatically determine the type and intensity of amino acid interactions among all possible mutational combinations for the three mutations introduced (Supplementary Note 6).

We quantified all amino acid interactions among all 6 trajectories leading from parent --- to mutant III for multiple parameters

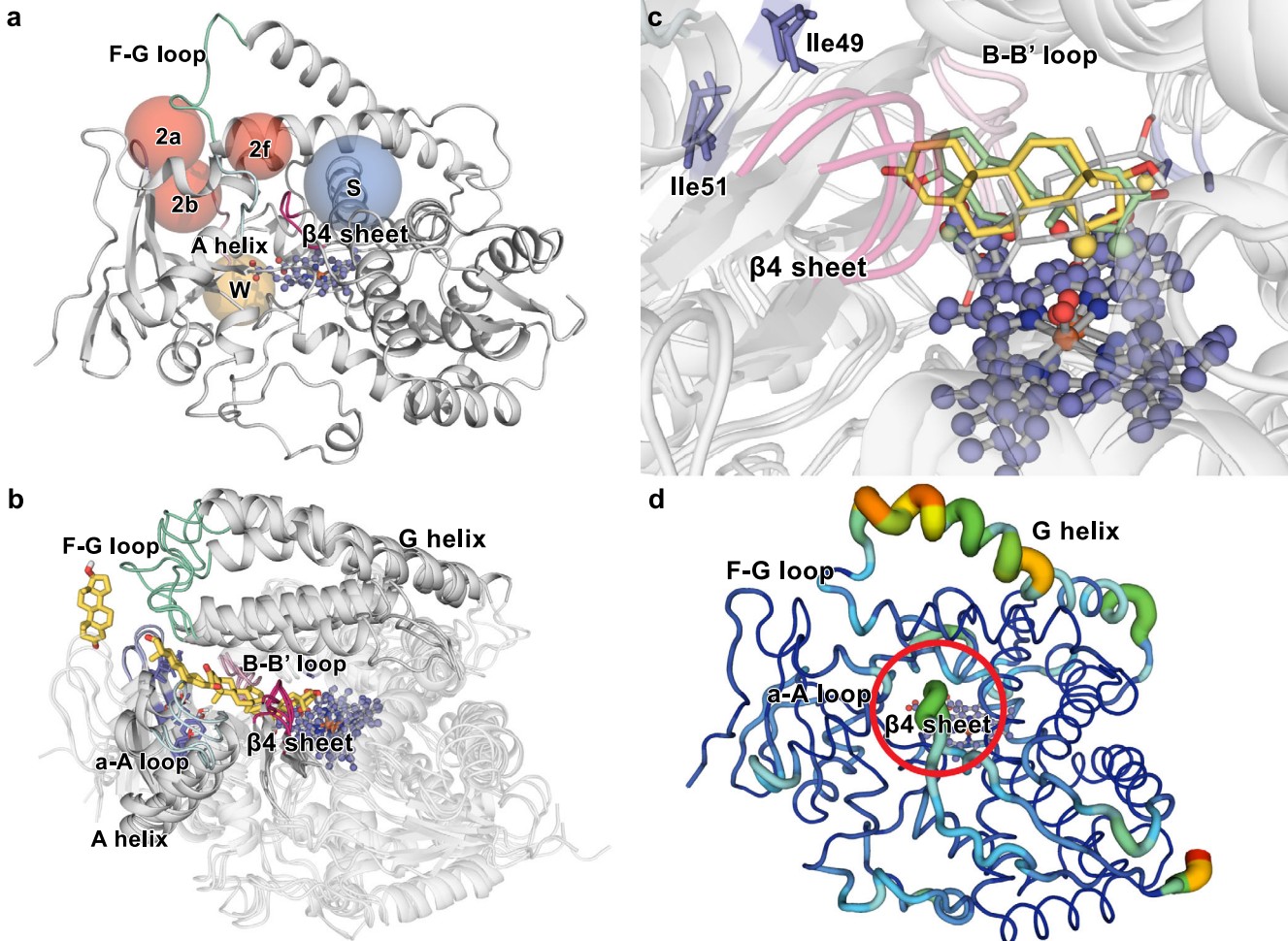

**Fig. 4 Secondary structural elements determining regioselectivity and activity. a** Spheres indicate the channels observed in the WT crystal structure (PDB: 1FAG) and in the mutants of our MD simulations (red and blue colour). **b** Trajectories of **1** towards the active site of mutant I-- and binding of **1** above the haem. **c** Rotation of **1** from pose15 to pose2 in the active site of mutant III. β Hydrogens belonging to C2 and C15 atoms are depicted in pale green and yellow colour, respectively. **d** Principal Component Analysis (pc2) of mutant III (APO). The thickness of the line is proportional to the motion and the colour scale varies from blue (minimum motion) to red (maximum motion). The β4 sheet is highlighted with a red circle. Standard nomenclature for channels[69] and secondary structure elements[70] is used.

focused on the evolution towards 2β-hydroxytestosterone (Table 1). All combinations on substrate conversion show synergistic effects, with 6 (86%) and 1 (14%) cases of SE and ME, respectively (Supplementary Table 3). For 2β-selectivity, all interactions are likewise synergistic, with most of them showing positive SE and only one case of positive ME (combination of R47I and T49I). For example, the combination of the single mutations R47I, T49I and Y51I (in parent mutant ---) is expected to contribute −3.45 ± 0.25 kJ/mol. The two former mutations confer 15β-selectivity in mutants I-- and -I-, while the latter one induces 2β-selectivity in variant --I. Yet the experimental value of mutant III yields 5.6 ± 0.0 kJ/mol, which represents a difference of about 9 kJ/mol between the experimental and theoretical values (Supplementary Table 4). Whereas NCR has 6 cases of positive epistatic effects (86%) and one negative case (14%), PFR likewise shows 6 cases of positive effects but one additive case (Supplementary Tables 5 and 6). Similarly, CE shows 6 cases of synergistic epistatic effects and 1 antagonistic case (Supplementary Table 7). Finally, TTF and TTN show the same synergistic effects of 70% SE and 30% ME (Supplementary Tables 8 and 9). Overall, these results indicate that an efficient consumption of NADPH and oxidation of testosterone towards formation of 2β-hydroxytestosterone requires pervasive

cooperative effects among R47, T49 and R51 regardless of mutational combination.

**Conformational dynamics shape the evolution of the fitness landscape.** The complete deconvolution of a multi-mutational variant enables the exploration of all possible pathways from parental enzyme to the evolved mutant, thus determining a full multidimensional fitness landscape. Such landscapes provide insights on the different routes that evolution can take. Additionally, engineering proteins by single mutational steps[53] is a highly successful strategy in directed evolution[2,4,5,38,54,55]. To explore the step-wise accessibility in the evolution of parental --- towards III, we constructed a fitness "pathway" landscape[56] based on both activity and selectivity (Fig. 6a). This system is a 4-dimensional surface (3 sets of mutations as independent vectors and ΔΔG‡ as the dependent variable obtained from the experimental selectivities). Two kinds of trajectories can be noted: those lacking local minima (favoured) and those characterized by at least one local minimum (disfavoured). Pathways 1–4 are characterized by a decrease in both selectivity and activity at the first step, indicating that they are evolutionarily disfavoured (pathway 3 is highlighted in red). Pathway 5 (highlighted in green) and 6

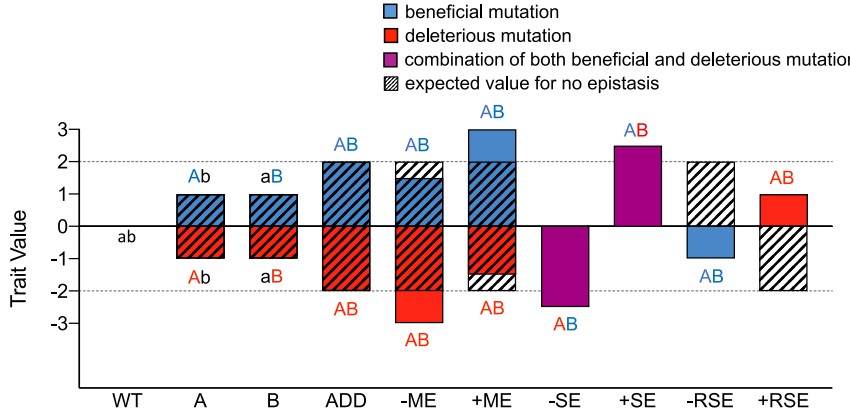

**Fig. 5 Explanation of non-additive effects in single mutations A and B.** Additivity (ADD) occurs when the sum of the individual effects of mutations A and B is equal to the value in double mutant AB (black diagonal lines). Epistatic effects emerge when the result of combining two individual mutations (or sets of mutations) is non-additive, i.e. the fitness value after combining mutations A with B to generate a doubly mutated AB variant is not equal to the sum of the individual A and B contributions. Epistatic effects can be positive/synergistic or negative/antagonistic depending upon their result with respect to additivity. They occur in the form of: (i) positive magnitude epistasis (+ME) if both the single mutations A and B are beneficial for a fitness trait and they produce a greater-than-additive fitness improvement (or a smaller-than-additive fitness drop if both are deleterious) when combined together in mutant AB; (ii) negative magnitude epistasis (−ME) if both the single mutations A and B are beneficial for a fitness trait and they produce a smaller-than-additive fitness improvement (or a greater-than-additive fitness drop if both are deleterious) when combined into AB; (iii) positive sign epistasis (+SE) if one mutation A is deleterious on its own but can enhance the beneficial effect of another mutation B when combined into AB; (iv) negative sign epistasis (−SE) if one mutation A is beneficial on its own but can enhance the deleterious effect of another mutation B when combined into AB; (v) positive reciprocal sign epistasis (+RSE) if both mutations A and B are deleterious alone, but they produce a beneficial effect when combined into AB and (vi) negative reciprocal sign epistasis (−RSE) if both are beneficial but they produce a deleterious effect when combined into AB.

**Table 1 Epistatic analysis of all possible mutational combinations on multiple parameters towards formation of product 2β-hydroxytestosterone.**

| Type[a] | Combination of mutations[b] | Resulting mutant[b] | Parameter[c,d,e] | | | | | | |
|---|---|---|---|---|---|---|---|---|---|
| | | | Conv. | Sel. | NCR | PFR | CE | TTN | TTF |
| B | I-- + -I- | II- | +SE 0.4 | +ME 2.8 | -SE −1.3 | +SE 1 | +RSE 3 | +SE 9 | +SE 2 |
| B | -I- + --I | -II | +SE 25 | +SE 5.6 | +ME 91 | +ME 39 | +SE 16 | +ME 490 | +ME 51 |
| B | I-- + --I | I-I | +SE 2 | +SE 2.6 | +SE 23 | ADD −0.2 | −SE −7 | +SE 38 | +SE 7 |
| B | I-- + -II | III | +SE 3 | +SE 3.5 | +SE 192 | +SE 70 | +SE 2 | +SE 58 | +SE 90 |
| B | II- + --I | III | +SE 27 | +SE 6.3 | +SE 284 | +ME 108 | +ME 15 | +SE 539 | +SE 139 |
| B | I-I + -I- | III | +ME 25 | +SE 6.4 | +ME 260 | +ME 109 | +SE 24 | +ME 509 | +ME 133 |
| T | I-- + -I- + --I | III | +SE 27 | +SE 9.0 | +SE 283 | +SE 109 | +SE 18 | +SE 548 | +SE 140 |

This shortened data set originates from Supplementary Tables 3–9.
[a]Binary (B) and tertiary (T) combinations.
[b]See Fig. 1c for nomenclature of mutations and mutants.
[c]The parameters (units) are: Conversion (Conv.), selectivity (Sel.), NADPH consumption rate (NCR), product formation rate (PFR), coupling efficiency (CE), total turnover number (TTN), total turnover frequency (TTF). The units can be found in Supplementary Tables 3–9.
[d]The types of epistatic effects are: Sign Epistasis (SE). Magnitude Epistasis (ME), Reciprocal Sign Epistasis (RSE), which can be positive (+) or negative (−), Additivity (ADD) means absence of epistatic effects.
[e]The data represent the average of two independent experiments ($n = 2$). The standard error mean can be found in Supplementary Tables 3–9.

are favoured because --I enables conformational changes in the active site and has implications on the substrate binding (as discussed above). In the two latter pathways, activity improves slightly in the evolution of --- towards --I (TTF = 11 → 21) at the first step, but at the second and third steps of pathway 6 it increases significantly towards -II and III (TTF = 21 → 72 → 158). This is due to the β4 sheet that shows an increased flexibility in the most active mutants -II and III, highlighting the key role of β4 sheet for activity (Fig. 6b). Interestingly, when all other parameters are considered, pathways 5 and 6 are the only ones that remain accessible (Supplementary Note 7 and Supplementary Fig. 14), with selectivity and CE showing the strongest non-additive effects, thus indicating the importance of residue Y51 towards efficient 2β-hydroxytestosterone formation.

To identify the most important conformational changes in all evolutionary pathways and to describe how distal mutations

influence them, we performed extensive MD simulations in the absence of substrate of each variant and applied the dimensionality reduction technique PCA[29] to the whole data set (Figs. 6c and 7a). A conformational population analysis resulting from all the accumulated simulation data was generated in terms of principal components (PCs) 1 and 3, which describe the first and third most important conformational differences among all variants (for PC2, see Supplementary Fig. 16). Notably, a clear distinction between 2β- and 15β-selective mutants is revealed through their separation with respect to PC1 (x-axis), suggesting that changes in selectivity are linked to the impact that the introduced mutations have on the enzyme conformational dynamics (Fig. 6c). These conformational changes related to selectivity mainly involve the G helix, the F–G loop, the β1 hairpin and the B' helix (located at the entrance of the 2a/b channels) as well as

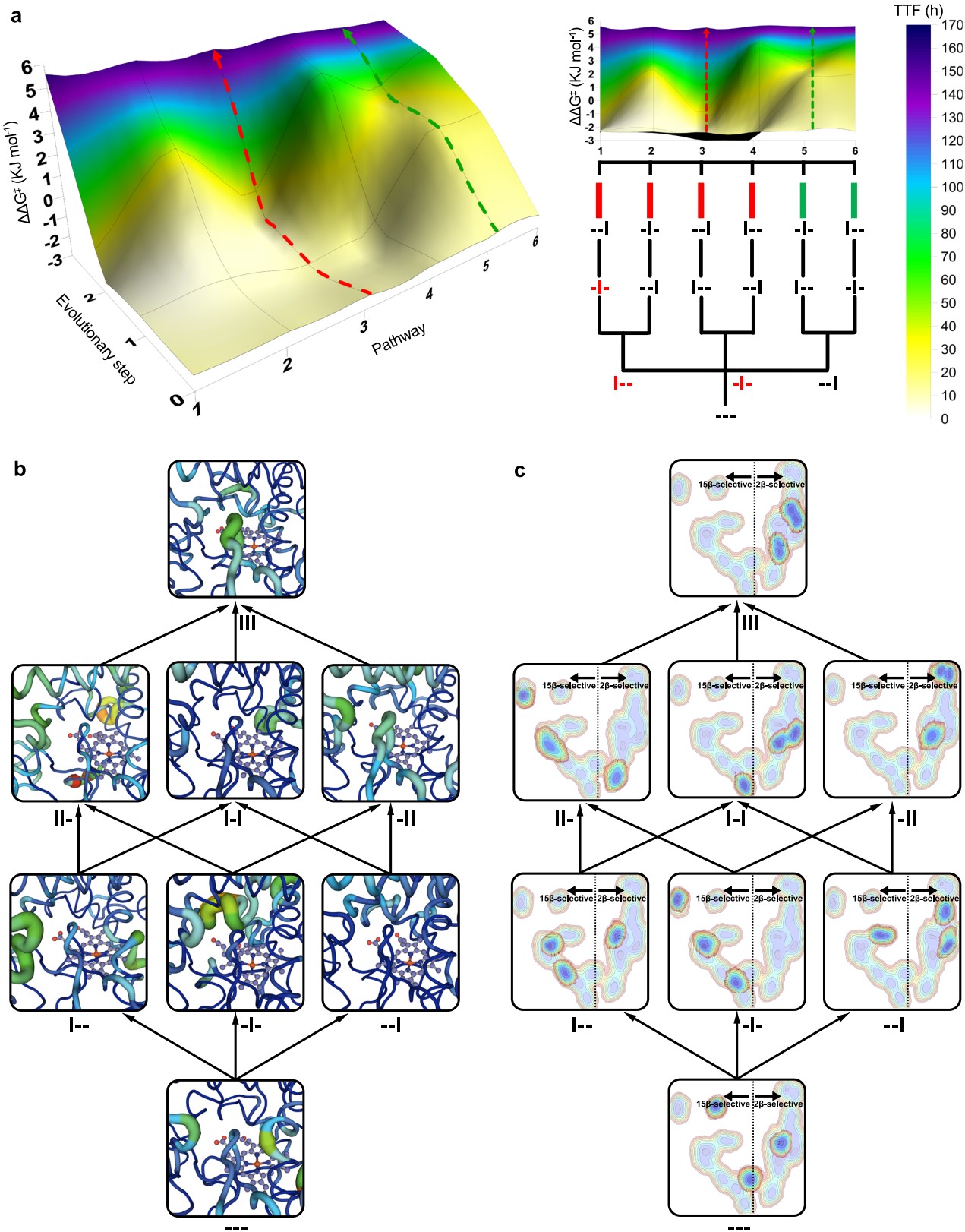

the a-A loop and the β4 sheet (located at the entrance of the 2f channel) (Fig. 7b)[57,58]. In variant -I-, the channel 2a has a narrower substrate access entrance due to a closed state of the F–G loop (ca. 9.3 Å determined between the Cα of R47 and N192). Conversely, the combination of mutations introduced in III favours an open conformational state of the same F–G loop (ca. 12.4 Å measured between the Cα of I47 and N192), enlarging the access channel 2a, which is mainly responsible for allowing access to the enzyme-binding pocket (Fig. 7c). Indeed, the area surrounding the access channel 2a in mutant III is calculated to have a volume of 140 Å$^3$ with respect to 44 Å$^3$ in -I-.

**Fig. 6 Stepwise evolution of multiple functions and conformational dynamics. a** Multiparametric fitness pathway landscapes of the 6 evolutionary pathways leading from parent mutant --- upward to mutant III are shown in 3D (left) and frontal (right) format (green and red arrows indicate examples of favourable and unfavourable pathways, respectively). Fitness is defined by activity as total turnover frequency (TTF) displayed as heat-maps from 0 (white) to 170 (blue) as well as 2β-selectivity as $\Delta\Delta G^{\ddagger}$ in kJ per mol (y-axis) at each evolutionary step (z-axis) for each pathway (x-axis). Green and red bars indicate favoured and disfavoured pathways, respectively. The mutant in red represents steps with disfavoured energy, i.e. the point where the pathway is blocked. The data represent the average of two independent experiments ($n = 2$). Original data are listed in Supplementary Table 1 and pathway analysis in Supplementary Tables 10 and 11. **b** Progression of the β4 sheet flexibility along the 6 pathways as revealed by PC analysis (pc2) of the substrate-free simulations of mutated enzymes analysed separately (see complete haem domains in Supplementary Fig. 15). The thickness of the line is proportional to the motion and the colour scale varies from blue (minimum motion) to red (maximum motion). **c** Evolution of the conformational dynamics along the 6 pathways and its connection to 2β- or 15β-selectivity. The analysis of the global conformational dynamics of the substrate-free simulations of mutated enzymes, as shown by pc1/pc3, indicate that 2β- and 15β-selective mutants explore conformations lying at positive and negative values of pc1, respectively. Colour scale varies from red (less populated) to blue (more populated). See Fig. 1d for mutant abbreviations.

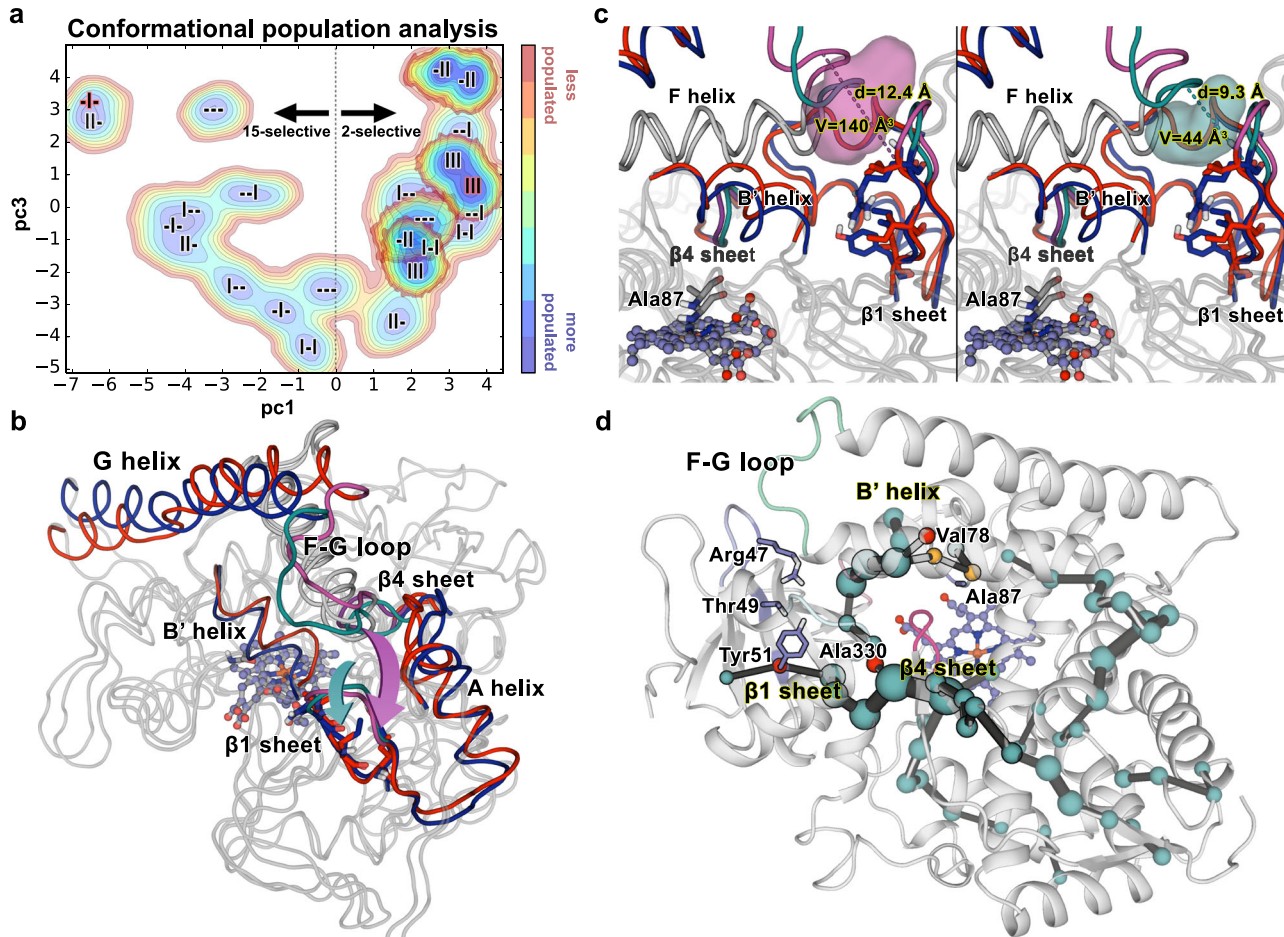

**Fig. 7 Analysis of the conformational dynamics of deconvolution mutants. a** Conformational population analysis built from the combined PCA of all substrate-free simulations of mutants. The conformational populations of 2β-selective mutants -II and III are highlighted. All replicas (3/3) of III and -II, 2/3 replica of I-I and --I lie on positive values of pc1 (10 out of 12 replicas of 2β-selective mutants). All replicas (3/3) of -I-, 2/3 replicas of ---, I-- and II- show negative values of pc1 (9 out of 12 replicas of all 15β-selective mutants). **b** Overlay of the conformational changes involved in PC1. **c** Zoom of **b** showing the channel 2a cavity of mutant III (left panel) and -I- (right panel). In **b, c**, Mutant -I- is shown in blue, whereas the evolved mutant III is coloured in red. The F-G loop, β1 sheet and channel 2a cavity are highlighted in teal and magenta for -I- and III, respectively. **d** Analysis of the most important correlated motions of mutant --- by means of the shortest path map (SPM). Mutational hotspot used in this or precedent work[41] that appear in the SPM are highlighted with red spheres, whereas positions adjacent to mutational hotspot are highlighted with orange spheres. See Fig. 1d for mutant abbreviations.

To further study the link between epistasis and conformational dynamics, we applied the shortest path map (SPM) analysis[22] (Supplementary Note 9) using the accumulated 1.8 μs MD simulation performed on parent --- in the absence of substrate. SPM considers the different conformations that the enzyme samples along the MD simulation and identifies which residues are those that are more important for the observed

conformational changes, which in this case are associated with different selectivities and activities[22]. In the parent --- enzyme, the generated SPM identifies residues Y51 as well as V78 and A330, known from earlier studies[40], to be important for enzyme activity and to be interconnected in terms of Cα correlated movements, thus highly contributing to the enzyme inactive-to-active conformational interconversion. This highlights why these

three distal positions are found to be key during the evolutionary pathway for improving catalysis, in line with what we observed for the laboratory-evolved retro-aldolases[22]. Importantly, the SPM also describes strong connections between all the five-stranded β1 sheet with the β4-2 strand and the B' helix, which we showed to be crucial for substrate binding and gating (Fig. 7d). This long-distance communicating pathway between β1 and β4 sheets directly relates the mutated positions on β1-2 strand (positions R47, T49 and Y51) and the increased flexibility of the β4 sheet. This shows how evolutionary pathways take advantage of networks of residue–residue interactions to fine tune the conformational dynamics along the evolutionary pathways for improving enzyme function.

## Discussion

The identification of epistasis and its molecular mechanism are crucial for understanding protein function, but these are hardly ever explored in laboratory evolution studies of multiparametric optimization. For example, the directed evolution of activity and selectivity in plant sesquiterpene synthases[59] or in P450$_{BM3}$ (ref. [60]) did not consider epistatic effects, while "stability-mediated epistatic effects" were observed in a P450$_{BM3}$ study[61] some years later[5]. Conversely, various studies in evolutionary biology have determined the contribution of multiple parameters on epistasis and organismal fitness. Shakhnovich et al., for instance, found that bacterial growth depends on the activity and expression of adenylate kinase[62] or on the activity, binding and folding stability of dihydrofolate reductase[63].

The construction of fitness landscapes using a single catalytic parameter has been reported in two main research areas. While such landscapes have revealed that usually many pathways are accessible in laboratory evolution of enzymes as catalysts in organic chemistry[4,7,56,64], different conclusions have been made in evolutionary biology[8–11]. In the present study, we observed that only a few trajectories (2/6) are accessible to both selectivity and activity. Interestingly, the two accessible pathways for selectivity correspond to mutation Y51I. The addition of mutation R47I has almost no effect on activity and selectivity, while mutation T49I, which is closer to Y51I, significantly improves both parameters as it alters the enzyme conformational dynamics. T49I and Y51I enhance the flexibility of the β4 sheet, and both combined with R47I reshape the active site for enhanced 2β-hydroxylation. The triple mutant III excels in all parameters compared to all double and single mutants. Unexpectedly, upon going from the "parent" enzyme --- to mutant III, cooperative interactions at each step in the evolution of selectivity and activity (i.e. TTF) remain pervasive, with SE and ME characterizing those effects. Residues R47I, T49I and Y51I are located at the entrance of a long substrate channel far away from the active site[45]. Since the mutated residues were not observed to interact directly with the substrate in our MD simulations, we propose that the observed epistatic effects, which are mediated by long-range interactions, can occur via one main mechanism: direct effects between mutations but no direct interaction between the substrate and the mutations[11].

Our computational exploration of the mutation-induced conformational changes on F87A variants provide key insights concerning the importance for P450$_{BM3}$ evolution towards more active and selective variants. These simulations predict that activity is dictated by the flexibility of the β4 sheet, which acts as a gate and modulates substrate access to the catalytic haem pocket for efficient hydroxylation. Our simulations also highlight the key role of the F–G loop in open–close conformational transitions involved in substrate binding, as shown for P450$_{BM3}$ by Shaik[58] and for P450$_{PikC}$ by Houk and Sherman[65]. The substrate-binding simulations show that selectivity is dictated by how the substrate is oriented when accessing the haem pocket through the β4 sheet. By tuning the open/close conformational states of the F–G loop and the β1 hairpin, the substrate access channels are altered, which impact substrate orientation and thus selectivity. This rich conformational heterogeneity observed for P450$_{BM3}$, which is important for substrate binding, is in line with previous reports[58] and also with the selective stabilization of discrete conformational states of P450$_{CYP119}$ and P450$_{PikC}$ upon ligand binding[65,66]. However, our simulations contrast to what was previously observed in P450$_{cam}$, which does not depend on open/closed conformational changes of the F, G helices and loop for allowing substrate binding[49]. It should also be mentioned that P450$_{cam}$ complexed with its redox partner adopts an open conformation that stabilizes the active site key for the proton relay network[67].

Using SPM analysis, the most important positions that participate in the open/closed conformational conversions that dictate selectivity and activity were predicted. Of relevance is that the key residue Y51 found to be essential for both activity and selectivity in this study is contained in the SPM path, as well as the previously described V78 and A330 positions[40]. SPM also highlights a long-distance communicating pathway between β4 and β1 where positions R47, T49 and Y51 are located, which is exploited along the evolutionary pathway for altering BM3 protein function.

This study provides evidence that in P450$_{BM3}$ epistasis is intrinsically linked to conformational dynamics, which fine-tunes multiple functions in a protein involved in secondary metabolism. Our findings on the conformational changes connected to CYP activity and selectivity and residue networks that modulate such conformational conversions can be expected to facilitate future rational evolution of these enzymes for diverse practical applications.

## Methods

**Chemicals, materials and software**. All commercial chemicals were purchased with the highest purity grade (e.g., high-performance liquid chromatography (HPLC)) from Sigma-Aldrich (St. Louis, US) unless otherwise indicated. For protein purification, lysozyme and DNase I was purchased from Applichem (Darmstadt, Germany). For PCRs, KOD Hot-Start DNA Polymerase was obtained from Novagen (Merck, Darmstadt, Germany). Restriction enzyme *DpnI* was bought from New England Biolabs (Ipswich, US). The *E. coli* BL21-Gold(DE3) strain, obtained from Novagen (Merck-Millipore) and generally cultured in lysogeny broth (LB) with 50 μg/mL kanamycin ($^{Kan50}$) as marker (LB$^{Kan50}$), both obtained from Carl Roth, was used for transformation of site-directed mutagenesis reactions as well as for protein overexpression experiments. According to standard molecular biology protocols, electro-competent *E. coli* cells were prepared using 10% glycerol (Applichem) and transformed with the corresponding plasmids using a "MicroPulser" electroporator (BioRad, Hercules, US) following the manufacturer's instructions. Oligonucleotides were purchased from Metabion (Martinsried, Germany). The analysis of sequencing reads was performed using the commercial software MegAlign from DNASTAR Lasergene version 11 (Madison, US) and the freeware ApE plasmid editor version 2.0.44 by Wayne Davis. The software used for constructing the fitness pathway landscapes is Surfer version 8 (Golden, US) and the graphs and dot plots were done with GraphPad Prism version 9 (La Jolla, US).

**Site-directed mutagenesis**. The mutants were created using the MegaPrimer method as reported in ref. [40]. Briefly, the P450$_{BM3}$ mutant F87A gene, already cloned in the pETM11-BM3 plasmid[40], was amplified by PCR by using <25 ng template with 2.5 μM of both silent and mutagenic oligos depending on mutant (Supplementary Table 12) in 50 μL of 1× KOD hot start buffer, 2 mM dNTPs (each), 25 mM MgSO$_4$, and 0.5 units of KOD hot start polymerase. The PCR programme started with 1 cycle of 95 °C for 3 min, 5 cycles of 95 °C for 30 s, 62 °C for 1 min, 72 °C for 6 min, 20 cycles of 95 °C for 3 min, 68 °C for 8.5 min, 1 cycle of 68 °C for 10 min and cooling. The samples were treated with 1 μL *DpnI* and incubated at 37 °C overnight to remove the parent plasmid. For each mutant, 5 colonies were incubated in 4 mL LB and the plasmids were extracted using the commercial kit QIAprep Spin Miniprep Kit from QIAGEN (Hildesheim, Germany). DNA sequencing was conducted with the four respective oligos listed in Supplementary Table 12 by service provider GATC (now Eurofins, Constance, Germany).

**$P450_{BM3}$-based oxidation reactions using purified enzymes**. Biotransformation reactions were performed as follows[38]. Briefly, reactions were performed in 2.2 mL microtitre plate (MTP) format by resuspending the thawed cells in 600 µL of reaction mixture, followed by addition of 6 µL testosterone [stock: 100 mM (dimethylformamide (DMF)); final conc. 1 mM (1%)], plate sealing with Breathe Easier sealing membranes (Sigma) and incubation in an orbital shaker with tray with holders for 96-well plates (Multitron, Infors HT, Switzerland) at 220 rpm and 25 °C for 24 h. The reaction mixture consisted of 100 mM KPi buffer pH 8.0, 100 mM glucose (Applichem), 10% glycerol (Applichem), 1 mM NADP$^+$ (Merck-Millipore or Applichem), 1 U/mL glucose dehydrogenase (GDH-105) obtained from Codexis (Redwood City, US), 5 mM EDTA and 50 µg/mL kanamycin. The reaction was stopped by adding 350 µL of ethyl acetate using a Tecan robotic system (Männedorf, Switzerland) equipped with a liquid handling arm (LiHA), which was controlled using the Gemini software V3.50, followed by centrifugation (10 min, 1100 × $g$, 4 °C). The organic phase was extracted using the same robotic system but with the multi-pipette option (Te-MO), transferred to 500 µL MTPs (Nunc, Roskilde, Denmark) and left unsealed for evaporation in the fume hood overnight. The dried samples were resuspended in 150 µL acetonitrile and passed through a PTSF 96-well plate filter to remove solid particles (Pall, VWR, Germany) into a new 500 µL MTP (Nunc). The MTPs, which were closed using silicon lids for the corresponding plates, were stored at 4 °C prior to screening.

**Steroid hydroxylation screening by HPLC**. A LC-2010 HPLC system (Shimadzu, Japan) equipped with four MTP racks was used employing a reverse-phase "250 Eclipse XDB" C18 column of 250 mm (1.8 µM size particle) together with a corresponding pre-column bought from Agilent (Waldbronn, Germany) as stationary phase and installed in the oven at 40 °C. The mobile phase was composed of a mixture of high-purity water generated from the local deionized water supply using a TKA MicroLab water purification system, acetonitrile (CH$_3$CN) and methanol (MeOH). For testosterone (**1**), a programme of 8 min based on a CH$_3$CN:MeOH:H$_2$O mixture was used: 0 → 3 min (15:15:70), 3 → 5 min (20:20:60), 5 → 6 min (30:30:40), 6 → 7 min (15:15:70). This protocol allows the separation of >14 oxidation products of **1**. The retention times of the known and unknown compounds can be found elsewhere[38]. Data acquisition was done using the Shimadzu LCsolution software version 3, while data analysis was performed with Microsoft Excel 365 MSO version 2012 (16.0.13530.20054) 32-bit.

**Large-scale protein expression and purification**. The $P450_{BM3}$ mutants were inoculated into 4 mL LB$^{Kan50}$ broth and cultured overnight in the orbital shaker with tray with adhesive matting for shake flasks (Multitron) at 37 °C and 220 rpm. The overnight culture (4 mL) was transferred into 200 mL TB$^{Kan50}$ in 500 mL shaking flasks. The cultivation continued at 37 °C and 220 rpm for 2–3 h until the OD$_{600}$ reached ~0.6–0.8, then IPTG was added to a final concentration of 100 µM and the temperature was reduced to 25 °C. After 20 h expression, the cells were harvested by centrifugation at 1100 × $g$ and 4 °C for 15 min. The cell pellets were stored at −80 °C until further processing. The cell pellets were dissolved in buffer (50 mM KPi, 800 mM NaCl, pH 7.5) and disrupted by sonication under an ice bath. The collected lysate was centrifuged for 45 min at 30,000 × $g$ at 4 °C and the obtained brownish-red supernatant was filtered to sterility with a 0.45-µm filter. The lysate obtained was loaded onto the pre-equilibrated nickel affinity column (HisTrap FF, 5 mL, GE Healthcare) with loading buffer (50 mM KPi, 800 mM NaCl, pH 7.5, 2 mM L-histidine). The column was first washed with 10 column volumes loading buffer, followed by gradient elution using an L-Histidine buffer (50 mM KPi, 80 mM L-histidine, pH 7.5) until complete protein elution. Columns were stripped and recharged between each mutant to avoid cross contamination. A flow rate of 5 mL/min was used and all fractions showing adsorption at 417 nm were collected. Proteins from the flow through were pooled and the buffer was exchanged to 25 mM KPi (pH 7.5) by ultrafiltration using a 50 kDa Amicon Ultra centrifugal filter (Merck-Millipore) and then concentrated to 5 mL. To remove the bound endogenous fatty acid, gravity-flow protein purification with Lipidex 1000 (Perkin Elmer) chromatography was conducted. Ten millilitres of Lipidex resin stored in methanol was used for column packing, which was subsequently washed with 10 column volumes of water and 10 column volumes of buffer (25 mM KPi, pH 7.5). After that, the protein was applied onto the column. The column was then capped to leave the protein in contact with resin at room temperature for 1 h, allowing hydrophobic compounds to bind to the resin. The protein was completely eluted from the Lipidex resin with buffer (25 mM KPi, pH 7.5) and the column was cleaned with at least 10 column volumes of methanol. The purified protein was pooled, and the buffer was exchanged to 100 mM KPi (pH 8.0) by ultrafiltration using a 50 kDa Amicon Ultra centrifugal filter, and then concentrated to 1 mL and stored at −80 °C for further use. An aliquot was thawed at room temperature and enzyme concentration was determined by CO difference spectrum analysis prior to usage. The enzyme concentration determined for all intermediate mutants is shown in Supplementary Table 1.

**Determination of kinetic parameters using isolated enzymes**. The kinetic experiments were performed using a JASCO V-650 spectrophotometer (JASCO International CO., LTD, Japan) equipped with a PAC-743 Peltier temperature

control unit and UV-Vis-NIR Spectra Manager software II. All assays were performed in 100 mM potassium phosphate buffer (pH 8.0) at 25 °C using quartz cuvettes adapted for magnetic stirring (900 rpm). NADPH consumption was determined by measuring NADPH depletion monitored at 340 nm ($\varepsilon = 6.22$ mM$^{-1}$ cm$^{-1}$). A concentration of 0.24 mM NADPH was used in the reaction mixture. Due to uncoupling reactions, where NADPH is consumed without substrate hydroxylation, the rates were calculated by subtracting the rate of NADPH consumption in the absence of substrate. Reactions containing 0.2 mM testosterone dissolved in DMF with a final solvent concentration of 1% (v/v) were started with addition of 100 nM $P450_{BM3}$ enzyme in a final volume of 1 mL and these were monitored until NADPH was completely depleted, as measured by no change in absorbance at 340 nm (completion of the reaction). Afterwards, the reaction mixture was immediately transferred into 96 MTPs and frozen at −20 °C. Reaction mixtures of 600 µL were taken and mixed with ethyl acetate (2 × 150 µL) with the LiHA of the Tecan robot platform (dispensing speed, 600 µL/s) The organic phase was extracted using the Te-MO multi-pipette option and transferred to 500 µL MTPs (Nunc, Roskilde, Denmark). The solvent was dried overnight, and the next day the steroid was resuspended in 150 µL acetonitrile and passed through a PTSF 96-well plate filter to remove particles (Pall, VWR, Germany) into a new 500 µL MTP (Nunc). The MTPs were stored at 4 °C prior to screening. The kinetic parameters are shown in Supplementary Table 1.

**Reporting summary**. Further information on research design is available in the Nature Research Reporting Summary linked to this article.

## Data availability
The authors declare that all data supporting the findings of this study are available within the paper and its supplementary information files. Source data are provided with this paper.

## Code availability
Computer code for determination of epistatic effects is available at GitHub under https://github.com/matteoferla/Epistasis_Calculator (https://doi.org/10.5281/zenodo.4423157). The Dynacomm computer code for the SPM (https://doi.org/10.1021/acscatal.7b02954) is available from the authors upon reasonable request.

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

## Acknowledgements

Support from the Max-Planck-Society and the LOEWE Research cluster SynChemBio is gratefully acknowledged. A.L. thanks the support from the National Key Research and Development Program of China (2019YFA0905000). This study was also supported in part by the European Research Council Horizon 2020 research and innovation pro-gramme (ERC-2015-StG-679001, to S.O.), Spanish MINECO (project PGC2018-102192-

B-I00, to S.O.; project PID2019-111300GA-I00, to M.G.-B.; and Juan de la Cierva - Incorporación fellowship IJCI-2017-33411, to M.G.-B.), UdG (predoctoral fellowship IFUdG2016, to L.D.), and Generalitat de Catalunya AGAUR (SGR-1707, to S.O.; and Beatriu de Pinós H2020 MSCA-Cofund 2018-BP-00204, to M.G.-B.). M.P.F. is supported by the Wellcome Trust [203141/Z/16/Z] and the NIHR Biomedical Research Centre Oxford.

## Author contributions

C.G.A.-R., S.H., S.O. and M.T.R. conceived the project. C.G.A.-R., A.L. and S.H. created and purified mutants and measured their activity. P.L. and M.F. wrote the Python code for automated calculation of epistatic effects. J.S. constructed and analysed the fitness pathway landscapes. L.D. performed the computational modelling and analysis with support and guidance from M.G.-B. and S.O. All authors wrote, revised and approved the manuscript.

## Funding

## Competing interests

The authors declare no competing interests.
