## [Peer Review File · Nature Communications]

REVIEWER COMMENTS

Reviewer #1 (Remarks to the Author):

In this paper, Reetz and coworkers have explored the link between long-range conformational dynamics, epistasis and evolvability, in an engineered cytochrome P450 monooxygenase. This is achieved using a broad range of combined computational analyses, including the development of an interesting new approach for predicting epistatic effects computationally. Overall, this is an excellent and well-written paper, which I think is a valuable contribution to the literature, provided that the authors address the comments below.

1. In the introduction, the authors discuss negative/positive epistasis, stating that “positive epistasis increases the evolution of new protein functions because it allows access to mutational pathways that avoid deleterious downfalls”. While this is on the one hand correct, that in positive epistasis you avoid deleterious downfalls, the need for epistasis in itself is limiting on protein evolution because it reduces the number of possible trajectories you can take in sequence space (if the presence of mutations is non-additive, the order becomes important, and prior mutations are necessary for subsequent mutations to show effect). See for example extensive discussion of epistasis in various papers by Tawfik. This needs to be commented on as well. Also, according to Tawfik (see e.g. <https://www.nature.com/articles/nature05385>) negative epistasis has been associated with a higher tolerance for mutations. Also, Tawfik has for example discussed sign epistasis and the link to conformational disorder / evolvability in the following paper: <https://pubmed.ncbi.nlm.nih.gov/26004540/>

2. I am being pedantic, but on the bottom of pg. 3, molecular dynamics simulations allow for reconstruction of part of rather than the full conformational landscape, please be specific.

3. Additionally, later in the same paragraph on pg. 3, not all systems will tune conformational dynamics and those that do tune conformational dynamics will not necessarily do it in the same way, some will amplify some will dampen dynamics, depending on the specifics of the system. I would not say that it is key for novel activity, but rather that it can play an important role in the emergence of novel activities in relevant systems. Please rephrase.

4. On pg. 12, it is curious (and also counter-intuitive) that the most active variant shows the highest flexibility of the $\beta 4$ sheet – based on its location in Fig. 4, one would expect greater flexibility of the sheet to also impact the flexibility of the B-B' loop, which in turn would lead to weaker substrate binding interactions, although this is just chemical intuition. Can the authors please clarify / quantify this link between flexibility of the sheet and activity in more detail, since it's counterintuitive? (it is mentioned later in the paper but just that the increased flexibility appears to link with high activity, not with an explanation why).

5. I would recommend changing the section header on pg. 16 to “the fitness landscape” singular, to be able to use the plural an exhaustive analysis of the role of conformational dynamics in the evolution of the fitness landscapes of multiple distinct enzyme groups needs to be performed in order to be able to generalize, which is not the case here. Rather, the results, while interesting, are confined to the present system. The same applies to the first sentence of the final paragraph of the conclusions, “This study provides evidence that in this

system...” would be a more accurate representation of the present work.

6. Supplementary Note 2: is there a particular reason the authors use B3LYP and not a more modern functional?

7. Supplementary Figure 8: I know the authors mention earlier, but it would be helpful to comment briefly in the legend as well how the active site volumes were calculated so the reader doesn't have to jump around the SI trying to find this.

Reviewer #2 (Remarks to the Author):

The authors perform an in-depth examination of a small but carefully selected set of variants of an enzyme to identify determinants of fitness when multiple mutations are combined. The methods used are state-of-the-art and interpretation of the results is well-supported. Deconvolution of the different pathways from the start-point to the top variant provides extremely interesting insights.

The paper is clearly written. The results should be more consistently situated within their explicit context: three sites were mutated, motions were predicted by computational work, and the conformational landscape was determined over 1.8 ns. Here are examples of statements that should be modulated:

Abstract: 'Here, we report the first multiparametric fitness landscape': computationally, and over 1.8 ns. Those two points should be clarified throughout the manuscript. This computational work is of the highest quality yet simulations remain predictive and should be presented in such terms. This recurs throughout the discussion, beginning at line 476. For instance 'These simulations have shown, for the first time, that activity is dictated by the flexibility of the β 4 sheet': it is more accurate to state that the simulations predict or are consistent with. Line 496: 'Using SPM analysis, the most important positions (...) were identified.' And so on.

Introduction: 'allow the reconstruction of the enzyme conformational landscape'. Again, the framework of 1.8 ns is a key point. MD was used which limits the landscape to relatively fast motions, in the context of landscapes. Slower motions are usually tracked experimentally or using normal modes. The context of more complete landscapes should be included here, such as <https://www.frontiersin.org/articles/10.3389/fmolb.2018.00115/full> and others.

The authors propose that 'the connection between conformational dynamics and epistasis has been limited to a few model enzymes (mainly beta-lactamases): addition of the work of Tokuriki/Jackson on phosphotriesterase and of Kern/Fraser on cyclophilin A, among others, would better reflect this broadening area of research and place the new results presented here in the context of a growing body of work on varied enzyme classes.

The paragraph running from line 251 to 258 is somewhat speculative. The results of the PCA show a correlation between flexibility of the β 4-sheet and activity as defined by "highest TTF, NADPH consumption rate and PFR numbers" but may not be ultimately "responsible (for) controlling substrate binding". Adoption of more cautious wording is recommended. Similarly, although it is plausible that "favouring a more efficient substrate binding in a catalytically competent pose increases enzyme TTF, while NADPH leak is reduced due to a more efficient interaction between the substrate and the catalytically active Fe=O species once

generated”, wording should be more cautious.

Lines 277-278: ‘a Python-based computational program to automatically determine the type and intensity of amino acid interactions among all possible mutational combinations’: specify ‘for the 3 mutations made’.

Minor points:

A basic definition of sign and magnitude epistasis in the introduction would allow the authors to rapidly reach non-experts (who may not get to Fig 5 and are unlikely to read the supplementary material).

Supplementary Note 4: Accelerated Molecular Dynamic Simulations: what is the driving force for the substrate to ‘diffuse freely until being spontaneously recognized by the enzyme surface and finally getting in the access channel’ instead of diffusing freely into the bulk solvent?

Although I appreciate the simplicity of Table 1, the authors should consider including the delta G values in the table: are the differences generally large or trivial? This big picture is hard to gather from the text (lines 295 to 315).

Line 125: Clarify phrasing of ‘followed by a fast C-O forming bond formation’

Lines 129-130: ‘Complete deconvolution of variant III starting from parental F87A entails 3! = 6 theoretically possible upward pathways’: clarify ‘possible upward’

Figure 2 legend: ‘...(f) normalises TTN by time after NADPH depletion’: until NADPH depletion?

Figure 3 legend, correct ‘red blue’.

Line 172: mainly include mainly

Lines 394-396: add reference(s) to support statement ‘These conformational changes mainly involve the G helix (...) (located at the entrance of the 2f channel) (Fig. 7b).’

Revise quality of English in Supplementary Note 1.

Supporting information line 150: correct to ‘restraints’.

Reviewer #3 (Remarks to the Author):

This study describes a fitness landscape for all combinations of three mutations in a cytochrome P450 that had been previously engineered to hydroxylate testosterone. They report results on the kinetic parameters of the catalysis as well as molecular dynamics simulations, and analyze the epistatic interactions between these mutations as well as the possible orders that these three mutations can be added while monotonically increasing quantities of interest.

My basic feeling is that this study is not of sufficient interest to a broad audience of scientists to be suitable for Nature Communications, although the detailed exploration of this particular

triple mutant and how these mutations alter conformational dynamics would be of interest to the narrower community that is more specifically interested in directed evolution of cytochrome P450.

This is essentially a case study, which although in principle could be interesting to a broad audience it depends on what specifically is revealed. Here the big picture story is quite simple in that there is one large effect mutation and then two other mutations have differing fine-tuning effects depending on whether they occur. The authors emphasize that they measure multiple enzymatic traits, but I did not feel that this added complexity came together to bring additional insight. I also did not find the epistasis analysis to be particularly compelling, e.g. listing the sign and type of epistatic interaction between mutations in Table 1. The accessible pathways analysis was also, I felt, not particularly interesting, since (1) directed evolution is not restricted to single mutants and (2) these were all done on a F87A background, which is not a natural background and so the relevance to natural evolution is limited. I also felt that the claims of novelty concerning measuring multiple traits in fitness landscapes was not completely accurate. Fitness landscape studies often measure multiple traits, e.g. studies of antibiotic resistance are often interested in collateral sensitivity and will make measurements of activities for multiple antibiotics, or to give another example the recent study of Li and Zhang 2018 measures the fitnesses of 23,000 yeast tRNA genotypes in 4 environments (as opposed to the 6 genotypes considered here).

Other comments:

- In Figure 5, do the divisions within the bar heights make sense in a rigorous fashion, especially in the SE column where the bar is split into uneven segments? It seems that the point of epistasis is that the double mutant fitness is not equal to a sum of individual effects and so it does not make sense to put it that way.

- I am confused by the abstract-level mention of the epistasis calculations. These seem quite routine.

Reviewer #4 (Remarks to the Author):

Epistasis within a protein is supposed if the combined effect of several mutations deviates from the sum predicted by adding their individual effects. The aim of this study was to analyse and to quantify effects induced by all mutational pathways possible during protein engineering of P450 BM3 for the selective 15b-hydroxylation of testosterone. To this end, eight possible mutants were produced, characterized regarding their activity and regioselectivity. A developed computational program has revealed pervasive positive epistatic effects. Quantum mechanics and MD simulations were applied to link protein epistasis and conformational changes in P450 BM3.

The manuscript provides new insights into the laboratory evolution of P450 BM3. The findings are of interest to the P450 community and the wider field.

Several questions remained open:

1. According to Fig. 2 and Supplementary Table 1, the mutants -II (T49I/Y51I/F87A) and III (R47I/T49I/Y51I/F87A) have very similar properties, except for initial rates and, correspondingly, TTF. C2-regioselectivity of these mutants is very high (85% and 91%, respectively). However, pose 15 was found highly unstable and the substrate rapidly rotates

to C2 close to the Cpd I only in R47I/T49I/Y51I/F87A. Does it mean that the mutation R47I has such strong effect on substrate flexibility in the binding pocket?

2. If the regioselectivity in all variants is determined by the orientation adopted by the substrate while accessing the active site, and only in the variant III the substrate rotates inside the active site bringing C15 close to the Cpd I, does it mean that the arginine at position 47 has a strong impact on the substrate orientation (or re-orientation) because it is the only difference between -II and III.

3. If no substrate rotation was observed during MD simulations of the C2-selective mutants, does it mean that starting pose 2 was used? Please comment on it. What happens if start with pose 2 during MD simulations with the mutant III?

4. According to aMD simulations, enzyme regioselectivity is determined by the orientation of the substrate when accessing the catalytic site during the second step of the binding pathway. How does it relate to the observed rotation of the substrate in the variant III? Please comment on it.

Hereafter, a detailed description of all the reviewers' comments together with a point-by-point response to the concerns is provided. All changes included in the manuscript have been highlighted in yellow.

Reviewer #1:

In this paper, Reetz and coworkers have explored the link between long-range conformational dynamics, epistasis and evolvability, in an engineered cytochrome P450 monooxygenase. This is achieved using a broad range of combined computational analyses, including the development of an interesting new approach for predicting epistatic effects computationally. Overall, this is an excellent and well-written paper, which I think is a valuable contribution to the literature, provided that the authors address the comments below.

1. In the introduction, the authors discuss negative/positive epistasis, stating that “positive epistasis increases the evolution of new protein functions because it allows access to mutational pathways that avoid deleterious downfalls”. While this is on the one hand correct, that in positive epistasis you avoid deleterious downfalls, the need for epistasis in itself is limiting on protein evolution because it reduces the number of possible trajectories you can take in sequence space (if the presence of mutations is non-additive, the order becomes important, and prior mutations are necessary for subsequent mutations to show effect). See for example extensive discussion of epistasis in various papers by Tawfik. This needs to be commented on as well. Also, according to Tawfik (see e.g. <https://www.nature.com/articles/nature05385>) negative epistasis has been associated with a higher tolerance for mutations. Also, Tawfik has for example discussed sign epistasis and the link to conformational disorder / evolvability in the following paper: <https://pubmed.ncbi.nlm.nih.gov/26004540/>

Answer: We thank the reviewer for recognizing the significance of our work, and for pointing out the works by Tawfik. It is correct that negative epistasis can lead to higher tolerance of mutations (first reference). We have added the following sentence to the introduction: “On the other hand, negative epistasis has been associated with a higher tolerance for mutations, which is important because this mutational robustness enables protein stability and evolution.”. Concerning the second reference, which was cited earlier as Ref. 32 and now as Ref. 35, we added the following sentence when mentioning the link of epistasis to protein dynamics in the introduction: “For example, negative sign epistasis between two distal mutations limited dynamics of active site loops mediating substrate accessibility in a β -lactamase³⁵.”.

2. I am being pedantic, but on the bottom of pg. 3, molecular dynamics simulations allow for reconstruction of part of rather than the full conformational landscape, please be specific.

Answer: We thank the referee for this comment as well, which is also related to a comment of reviewer 2 about NMR below. We have changed the above-mentioned sentence, which now reads: “Molecular dynamics (MD) simulations, which are highly complementary to NMR analyses³¹, allow the partial reconstruction of the enzyme conformational landscape, and how this is altered by mutations introduced by laboratory evolution.”

3. Additionally, later in the same paragraph on pg. 3, not all systems will tune conformational dynamics and those that do tune conformational dynamics will not necessarily do it in the same way, some will amplify some will dampen dynamics, depending on the specifics of the system. I would not say that it is key for novel activity, but rather that it can play an important role in the emergence of novel activities in relevant systems. Please rephrase.

Answer: Following the referee's suggestions, we have additionally rephrased the last sentence of the paragraph, which now reads: "Tuning the enzyme conformational dynamics can play an important role in the emergence of novel activities."

4. On pg. 12, it is curious (and also counter-intuitive) that the most active variant shows the highest flexibility of the $\beta 4$ sheet – based on its location in Fig. 4, one would expect greater flexibility of the sheet to also impact the flexibility of the B-B' loop, which in turn would lead to weaker substrate binding interactions, although this is just chemical intuition. Can the authors please clarify / quantify this link between flexibility of the sheet and activity in more detail, since it's counterintuitive? (it is mentioned later in the paper but just that the increased flexibility appears to link with high activity, not with an explanation why).

Answer: We thank referee 1 for this comment. Based on the aMD binding trajectory of testosterone in mutant I-- (Supplementary Video 2), we have seen that the retreat of the $\beta 4$ sheet (which acts as a gate) is necessary to allow substrate progression into the haem pocket. Based on our Principal Component (pc2) Analysis of the deconvoluted dataset (Supplementary Figure 15), we have observed that the $\beta 4$ sheet is rather rigid in all the mutants except in the most active ones (-II and III), in which it is indeed more flexible. A higher flexibility of the $\beta 4$ sheet may facilitate its retreatment, triggering the gating function for the progression of the substrate towards the haem pocket. We agree that such flexibility may have an impact on the backbone element that might be opposed to the $\beta 4$ sheet. Yet from the PCA analysis performed on variant III (Supplementary Fig. 12a), it appears that the increased flexibility of the $\beta 4$ sheet has no impact on the B' helix / B-B' loop, whereas in mutant -II it has only a minor impact on the flexibility on the B' helix, and no impact on the B-B' loop (Supplementary Fig. 12b). These findings can be supported by looking at the root-mean square fluctuation (RMSF) measured from the simulations in the absence of substrate: Mutants III and -II show higher RMSF values of the $\beta 4$ sheet region compared to F87A, whereas the RMSF values are rather comparable for all enzymes over the B-B' loop and B' helix region (Supplementary Fig. 13).

We have added two new Figures in the SI (Fig. S12, Fig. S13), and clarified this point in the main text, as follows: "Interestingly, the analysis of the most relevant conformational changes in each independent variant through Principal Component Analysis (PCA) predicts that the most active mutant III shows the highest flexibility of the $\beta 4$ sheet (Fig. 4d). This higher flexibility related to activity has, however, no impact on the B' helix and B-B' loop conformational dynamics (Supplementary Figs 12-13).".

5. I would recommend changing the section header on pg. 16 to "the fitness landscape" singular, to be able to use the plural an exhaustive analysis of the role of conformational dynamics in the evolution of the fitness landscapes of multiple distinct enzyme groups needs to be performed in order to be able to generalize, which is not the case here. Rather, the results, while interesting, are confined to the present system. The same applies to the first sentence of the final paragraph of the conclusions, "This study provides evidence that in this system..." would be a more accurate representation of the present work.

Answer: Following the referee's comments, we have changed the section header on page 16 and the above-mentioned sentence in the discussion section.

6. Supplementary Note 2: is there a particular reason the authors use B3LYP and not a more modern functional?

Answer: The main purpose of these QM calculations was to model the ideal transition state geometry to estimate the ideal geometric features (distances and angles) for effective hydroxylation, and estimate whether these catalytically competent geometric constraints (in terms of distance and angle) were sampled along the MD simulations performed. The selection of B3LYP-D3 was based on the reasonably good geometrical estimations and barrier heights provided by this classic but still widely used functional. Indeed, B3LYP is still the functional used in many P450 studies and it has been shown that it performs well in predicting experimental

selectivity trends (check for instance: [j.jmgm.2014.06.002](https://doi.org/10.1093/jmgm/2014.06.002) about B3LYP-D3 accuracy in P450 haem-based calculations).

7. Supplementary Figure 8: I know the authors mention earlier, but it would be helpful to comment briefly in the legend as well how the active site volumes were calculated so the reader doesn't have to jump around the SI trying to find this.

Answer: We have added this information in the figure which has been renamed now as Supplementary Fig. 9.

Reviewer #2:

The authors perform an in-depth examination of a small but carefully selected set of variants of an enzyme to identify determinants of fitness when multiple mutations are combined. The methods used are state-of-the-art and interpretation of the results is well-supported. Deconvolution of the different pathways from the start-point to the top variant provides extremely interesting insights.

Answer: We thank the reviewer for this positive assessment.

The paper is clearly written. The results should be more consistently situated within their explicit context: three sites were mutated, motions were predicted by computational work, and the conformational landscape was determined over 1.8 ns. Here are examples of statements that should be modulated:

Abstract: 'Here, we report the first multiparametric fitness landscape': computationally, and over 1.8 ns. Those two points should be clarified throughout the manuscript. This computational work is of the highest quality yet simulations remain predictive and should be presented in such terms. This recurs throughout the discussion, beginning at line 476. For instance 'These simulations have shown, for the first time, that activity is dictated by the flexibility of the β 4 sheet': it is more accurate to state that the simulations predict or are consistent with. Line 496: 'Using SPM analysis, the most important positions (...) were identified.' And so on.

Introduction: 'allow the reconstruction of the enzyme conformational landscape'. Again, the framework of 1.8 ns is a key point. MD was used which limits the landscape to relatively fast motions, in the context of landscapes. Slower motions are usually tracked experimentally or using normal modes. The context of more complete landscapes should be included here, such as <https://www.frontiersin.org/articles/10.3389/fmolb.2018.00115/full> and others.

Answer: We agree with referee 2 that molecular dynamics (MD) simulations are highly complementary to other experimental techniques (such as NMR) that can elucidate conformational changes taking place in the slowest timescales. Indeed, MD can easily describe fast transitions that usually correspond to side-chain conformational changes and occur in the picosecond to microsecond timescales. However, many studies based on MD simulations have shown how slower processes such as loop motions (usually key for substrate binding, as shown in this study and in others) that occur in the nanosecond up to millisecond timescales can be successfully characterized. In fact, MD simulations, especially if multiple replicas are statistically combined or enhanced sampling techniques are applied, have been extensively used to characterize protein conformational landscapes (see for instance: [eLife.45403.001](https://doi.org/10.1093/eLife/45403)). We would like also to stress that the MD simulations performed in this study are over 1.8 microsecond rather than nanosecond as stated by the referee. We agree that although we focus our analysis on rather long (and multiple replica) MD simulations, the accessed simulation time does not allow the complete reconstruction of the enzyme conformational landscape, but they indeed provide relevant insights on how the introduced mutations alter the conformational ensemble and its connection with the experimentally observed epistatic effects.

We have changed some of the above-mentioned sentences and included the suggested reference, which now read as:

"Molecular dynamics (MD) simulations, which are highly complementary to NMR analyses³¹, allow the partial reconstruction of the enzyme conformational landscape, and how this is altered by mutations introduced by laboratory evolution."

“These simulations **predict** that *activity is dictated by the flexibility of the β 4 sheet*, which acts as a gate and modulates substrate access to the catalytic haem pocket for efficient hydroxylation.”

“Using SPM analysis, the most important positions that participate in the open/closed conformational conversions that dictate selectivity and activity were **predicted**.”

The authors propose that ‘the connection between conformational dynamics and epistasis has been limited to a few model enzymes (mainly beta-lactamases): addition of the work of Tokuriki/Jackson on phosphotriesterase and of Kern/Fraser on cyclophilin A, among others, would better reflect this broadening area of research and place the new results presented here in the context of a growing body of work on varied enzyme classes.

Answer: We thank you for pointing out those interesting works. We have added them accordingly: “The connection between conformational dynamics and epistasis has been studied in proline isomerase (cyclophilin A)²⁴, phosphotriesterase³⁴ and β -lactamases³⁵⁻³⁷.”

The paragraph running from line 251 to 258 is somewhat speculative. The results of the PCA show a correlation between flexibility of the β 4-sheet and activity as defined by “highest TTF, NADPH consumption rate and PFR numbers” but may not be ultimately “responsible (for) controlling substrate binding”. Adoption of more cautious wording is recommended. Similarly, although it is plausible that “favouring a more efficient substrate binding in a catalytically competent pose increases enzyme TTF, while NADPH leak is reduced due to a more efficient interaction between the substrate and the catalytically active Fe=O species once generated”, wording should be more cautious.

Answer: As mentioned by the referee, the performed PCA analysis for the different variants predicts a higher flexibility of the β 4-sheet in the case of the most active mutants. In addition to that, aMD simulations indicate that a 2-step binding choreography of the substrate (TES) occurs: (1) entering channel 2a and forming a long-lived substrate-enzyme bound intermediate, followed by (2) a coupled network of conformational changes that promote the retreat of β 4-sheet and allow TES access to the haem binding pocket (see also Supplementary Video 2). All together, these simulations predict that β 4-sheet flexibility plays a key role in regulating substrate access to the catalytic pocket. These findings are in line with the highest activity observed experimentally for the mutants exhibiting the largest β 4-sheet flexibility. We have, however, changed some sentences in the new revised manuscript, which now read as:

“Interestingly, the analysis of the most relevant conformational changes in each independent variant through Principal Component Analysis (PCA) **predicts** that the most active mutant III shows the highest flexibility of the β 4 sheet (Fig. 4d). These flexible regions, responsible of controlling substrate binding as described above, **are likely to** influence activity, as mutant III shows the highest TTF, NADPH consumption rate and PFR numbers.”

“**These findings suggest that** favouring a more efficient substrate binding in a catalytically competent pose increases enzyme TTF, while NADPH leak is reduced due to a more efficient interaction between the substrate and the catalytically active Fe=O species once generated.”

Lines 277-278: ‘a Python-based computational program to automatically determine the type and intensity of amino acid interactions among all possible mutational combinations’: specify ‘for the 3 mutations made’.

Answer: We have added “**for the 3 mutations introduced**”. Since we wrote a program that could be used to compute epistatic effects in any other protein and parameter (or catalytic trait for enzymes), we have modified slightly the following sentence in the introduction: “To determine epistatic effects effectively, we developed a Python-based script and freely-accessible web-app (<https://epistasis.mutanalyst.com/>), which can be used for any enzyme and catalytic trait (or for any protein and parameter).”

Minor points:

A basic definition of sign and magnitude epistasis in the introduction would allow the authors to rapidly reach non-experts (who may not get to Fig 5 and are unlikely to read the supplementary material).

Answer: We agree and have added the following sentence in the introduction shortly after introducing the definition of epistasis: “Sign epistasis occurs when a mutation has a deleterious or beneficial effect alone, but an opposite effect when combined with other(s) mutation(s), whereas in magnitude epistasis a mutation has a deleterious or beneficial effect in isolation and in combination with other mutations.”.

In addition, since we think that it is important to differentiate between these 2 forms of epistasis, we added the following sentence in the second paragraph of the discussion: “with sign and magnitude epistasis characterizing those effects”.

Supplementary Note 4: Accelerated Molecular Dynamic Simulations: what is the driving force for the substrate to diffuse freely until being spontaneously recognized by the enzyme surface and finally getting in the access channel instead of diffusing freely into the bulk solvent?

Answer: As stated in Supplementary Note 4, accelerated Molecular Dynamics simulations (aMD) enhance the conformational sampling by adding a bias potential (i.e. boost potential), which raises energy minima while keeping high energy regions almost unaffected. This smoothens the free energy landscape and enhances the sampling of rather slow conformational exchanges, thus allowing the exploration of conformational states playing a role in substrate binding. In the computational protocol used, we started with 250 ns of conventional Molecular Dynamics (cMD) simulations followed by 750 ns of aMD, which allowed the reconstruction of the substrate binding choreography from the bulk solvent to the haem binding pocket shown in Supplementary Video 2. We have added a few sentences in Supplementary Note 4 to better clarify this point.

Although I appreciate the simplicity of Table 1, the authors should consider including the delta G values in the table: are the differences generally large or trivial? This big picture is hard to gather from the text (lines 295 to 315).

Answer: Thank you for the comment. The delta G values (kJ per mol) as well as delta conversion (%), μ moles, and rates, which are listed in Supplementary Tables 3-9, have been added to Table 1. A note was added to Table 1 regarding the % of error. It is possible now to see that the differences. For example, in the case of selectivity, the lowest and highest values are 2.6 and 9 kJ per mol, respectively, which are large differences.

Line 125: Clarify phrasing of ‘followed by a fast C-O forming bond formation’

Answer: We have edited the sentence, which now reads: “followed by a fast C-O bond formation”

Lines 129-130: ‘Complete deconvolution of variant III starting from parental F87A entails $3! = 6$ theoretically possible upward pathways’: clarify ‘possible upward’

Answer: We removed the two words and left only “6 theoretical pathways”.

Figure 2 legend: ‘...(f) normalises TTN by time after NADPH depletion’: until NADPH depletion?

Answer: Corrected.

Figure 3 legend, correct ‘red blue’.

Answer: Corrected.

Line 172: mainly include mainly

Answer: Corrected.

Lines 394-396: add reference(s) to support statement ‘These conformational changes mainly involve the G helix (...) (located at the entrance of the 2f channel) (Fig. 7b).’

Answer: The role of conformational changes of the F and G helices, F-G loop and their impact on substrate specificity and binding have been highlighted in different computational and experimental studies (see for

instance: jbc.RA119.010352 and references therein). Shaik and co-workers showed the role of β 1 sheet and A helix in the substrate binding choreography (acs.accounts.8b00467 and references therein). However, our study is the first one describing the importance of β 4 sheet conformational changes for substrate binding in BM3. In this new revised version of the manuscript we have added the above-mentioned references as suggested by the referee.

Revise quality of English in Supplementary Note 1.

Answer: It has been revised and rewritten as follows: "Cytochrome P450s catalyze complex reactions involving many components such as substrate [S], product [P], cofactor (NADPH), and oxygen (O₂). These multicomponent enzymes exhibit atypical or so-called non-Michaelis Menten kinetics.¹ Steady state kinetics parameters (e.g., kcat/Km) may lead to inaccurate interpretations on enzyme performance because these do not consider the effects of changing substrate concentration and product inhibition during the biocatalytic reaction². Consequently, in order to assess the catalytic efficiency of the P450_{BM3} variants, we determined the following parameters:"

Supporting information line 150: correct to 'restraints'.

Answer: Corrected.

Reviewer #3:

This study describes a fitness landscape for all combinations of three mutations in a cytochrome P450 that had been previously engineered to hydroxylate testosterone. They report results on the kinetic parameters of the catalysis as well as molecular dynamics simulations, and analyze the epistatic interactions between these mutations as well as the possible orders that these three mutations can be added while monotonically increasing quantities of interest.

My basic feeling is that this study is not of sufficient interest to a broad audience of scientists to be suitable for Nature Communications, although the detailed exploration of this particular triple mutant and how these mutations alter conformational dynamics would be of interest to the narrower community that is more specifically interested in directed evolution of cytochrome P450.

This is essentially a case study, which although in principle could be interesting to a broad audience it depends on what specifically is revealed. Here the big picture story is quite simple in that there is one large effect mutation and then two other mutations have differing fine-tuning effects depending on whether they occur. The authors emphasize that they measure multiple enzymatic traits, but I did not feel that this added complexity came together to bring additional insight. I also did not find the epistasis analysis to be particularly compelling, e.g. listing the sign and type of epistatic interaction between mutations in Table 1. The accessible pathways analysis was also, I felt, not particularly interesting, since (1) directed evolution is not restricted to single mutants and (2) these were all done on a F87A background, which is not a natural background and so the relevance to natural evolution is limited. I also felt that the claims of novelty concerning measuring multiple traits in fitness landscapes was not completely accurate. Fitness landscape studies often measure multiple traits, e.g. studies of antibiotic resistance are often interested in collateral sensitivity and will make measurements of activities for multiple antibiotics, or to give another example the recent study of Li and Zhang 2018 measures the fitnesses of 23,000 yeast tRNA genotypes in 4 environments (as opposed to the 6 genotypes considered here).

Answer: We disagree with the general conclusion of the reviewer that our paper is of narrow interest. How mutations influence conformational dynamics, which the reviewer considers important, are always studied by a specific example of a protein/enzyme. Our work provides a deep understanding of the relationship between protein dynamics and epistasis by using high-level computations in a model cytochrome P450 that was evolved by laboratory evolution. In fact, this is the first time that a P450 enzyme has been deconvoluted to determine epistatic effects and to connect these with protein dynamics. Given the importance of cytochrome P450s in basic and applied research, we believe that our work is original and important. This is supported by the other 3 reviewers. To better reflect the importance of the model system employed, we changed the previous sentence "The haem-containing CYP super family is involved in the biosynthesis and degradation of organic molecules

in a wide range of secondary metabolic and thus have many applications in biocatalysis^{40,41.}” to the following in the first paragraph of the results: “The CYPs super protein family has over 300,000 members that are involved in the biosynthesis of steroids, fatty acids and natural products, among others, as well as in the degradation of drugs in humans and of xenobiotics in the environment.^{41,42} Thus, this is an important enzyme class with relevant applications in biocatalysis, biomedicine, pharmacology, toxicology and biotechnology.^{42-44.”}

It is true that one mutation out of three has a large effect on one protein trait (selectivity); however, the reviewer has refused to accept our conclusion that all three mutations acting in concert are necessary for the significant increase in catalytic performance. The evolution of two or more parameters is something rarely studied, and it can provide practical lessons in directed evolution. To exemplify this, we added the following paragraph shortly before Table 1: “These findings suggest that P450 engineering for steroid hydroxylation can focus first on selectivity and then on activity and not the other way around, because it is more challenging to overcome sign epistasis than magnitude epistasis.”

The reviewer “did not find the epistasis analysis to be particularly compelling, e.g. listing the sign and type of epistatic interaction between mutations in Table 1.” The epistatic analysis that we present is not trivial (see below our reply to the last comment of the reviewer). Studies of epistatic effects on multiple protein traits are scarce. For example, when assessing beta-lactamases in vivo with high-throughput screenings (see below), usually only activity is considered, but there is little understanding about other important parameters such as protein expression, stability, and degradation. As mentioned in the discussion, only a few studies have determined epistatic effects on multiple protein traits (Ref. 65-66 by Shakhnovich and colleagues). Note that multiple parameters (e.g. activity, selectivity, stability, binding, etc.) do not equate to “multiple environments”, although the latter would influence the former.

Many parameters listed in Table 1 relate to selectivity as well as activity related to the substrate (TTF, TTN, PFR, conversion) and cofactor (NADPH consumption rate). Table 1 shows that non-additive effects prevail for all parameters. While this is general for the P450 mutants tested, we have not concluded that this will be the case for other engineered or natural P450s. Certainly, more examples are needed including different P450 enzymes and substrates as well as parameters to better understand epistasis and dynamics in this important class of enzymes. Moreover, our study could inspire others to explore other protein classes, for which there are many that have not been studied in detail.

The reviewer also argues that:

a) “(1) directed evolution is not restricted to single mutants”. Of course, directed evolution can be done by combining many mutations to generate multi-mutational variants or via single mutational steps, which are preferred for some types of enzymes (see review by Tracewell and Arnold; doi: [10.1016/j.cbpa.2009.01.017](https://doi.org/10.1016/j.cbpa.2009.01.017)). In fact, dozens of studies have employed stepwise site-saturation mutagenesis to improve one or more protein properties, often simultaneously. To reflect more the importance of single mutations in directed evolution, we added the following sentence to the revised text in the beginning of the section of Figure 6: “Such landscapes provide insights on the different routes that evolution can take. Additionally, engineering proteins by single mutational steps⁵⁶ is a highly successful strategy in directed evolution^{2,4,5,38,57,58.”}

b) “(2) these were all done on a F87A background, which is not a natural background and so the relevance to natural evolution is limited.” In general, we think that the findings of directed evolution studies including epistasis should not be extrapolated to natural evolution. This has been discussed in earlier works (see Ref. 7). What happens in the test tube is a controlled experiment that should not be compared to what happens in situ, which involves different regulatory mechanisms in the presence of many non-controllable factors (e.g., lack of and competition for nutrients, influences on gene and protein expression, mRNA and protein stability and degradation, etc.). Thus, the above comment is not relevant to our study. Of course, there are other studies in which a single protein can determine organismal fitness (e.g., beta-lactamases or essential proteins), but this is generally not the case for cytochrome P450s. To better illustrate this aspect of our work, we changed the original first paragraph of the discussion:

“The identification of epistasis and its molecular mechanism are crucial for understanding protein function, but these are hardly ever explored in directed or natural evolution studies of multiparametric optimization. To

the best of our knowledge, only the study by Hartl and colleagues on fitness landscapes of drug resistance towards trimethoprim fits such a description, in which activity, binding and folding stability were shown to depend to some extent on epistasis⁶³. Other studies dealing with directed evolution of activity and selectivity in plant sesquiterpene synthases⁶⁴ and P450_{BM3}⁶⁵ do not consider epistatic effects, while “stability-mediated epistatic effects” were observed in a P450_{BM3} study⁶⁶ some years later⁵.”

for the following sentence: “The identification of epistasis and its molecular mechanism are crucial for understanding protein function, but these are hardly ever explored in laboratory evolution studies of multiparametric optimization. For example, the directed evolution of activity and selectivity in plant sesquiterpene synthases⁶² and P450_{BM3} (Ref.⁶³) did not consider epistatic effects, while “stability-mediated epistatic effects” were observed in a P450_{BM3} study⁶⁴ some years later⁵. Conversely, various natural evolution studies have determined the contribution of multiple parameters on epistasis and organismal fitness. Shakhnovich et al., for instance, found that bacterial growth depends on the activity and expression of adenylate kinase⁶⁵ or on the activity, binding, and folding stability of dihydrofolate reductase⁶⁶.”

c) “Fitness landscape studies often measure multiple traits, e.g. studies of antibiotic resistance are often interested in collateral sensitivity and will make measurements of activities for multiple antibiotics” This is true for enzymes such as beta-lactamases that can benefit from a high-throughput selection systems and deep-sequencing. Changing antibiotics (e.g., [doi: 10.1093/molbev/msv059](https://doi.org/10.1093/molbev/msv059)), environments (e.g., [doi: 10.1126/sciadv.1500921](https://doi.org/10.1126/sciadv.1500921)) or assay conditions (e.g., [doi: 10.1093/nar/gku511](https://doi.org/10.1093/nar/gku511)) obviously changes the fitness landscape and the values of the parameters under study. However, this does not equate to the determination of different protein parameters beyond activity. Moreover, in our case we used low-throughput HPLC screening, which is a limitation to study more genotypes and environments. Thus, this comment is not relevant to our study.

Finally, the work from Li and Zhang ([doi: 10.1038/s41559-018-0549-8](https://doi.org/10.1038/s41559-018-0549-8)) uses in vivo selection and deep sequencing of a large library (23,000 variants) of a tiny non-coding protein gene of 72 bases (tRNA) expressed in yeast cultured at 4 different temperatures. The work is based on a previous study from the same authors (Ref. 9) in which high-order epistatic effects were quantified in yeast using the same tRNA molecule. The only novelty of the study is the correlation between genotype, phenotype, and environment. However, as indicated above, environment does not equate with protein or tRNA parameter (e.g., folding). Unfortunately, the Li and Zhang papers refer only to the epistatic effects of the tRNA molecule and the fitness of yeast, and no computational work is given for any protein (e.g., aminoacyl tRNA synthetase). Thus, we do not believe that the work of Li and Zhang is directly relevant to our study.

Hopefully, now with the changes/additions in the revision, the reviewer will appreciate the significance of our contribution to a central question in enzymology.

Other comments:

- In Figure 5, do the divisions within the bar heights make sense in a rigorous fashion, especially in the SE column where the bar is split into uneven segments? It seems that the point of epistasis is that the double mutant fitness is not equal to a sum of individual effects and so it does not make sense to put it that way.

Answer: We appreciate this comment. It is true that the value of the double mutant should have no expected values of the individual mutations. For this reason, we have changed Figure 5. Hopefully, our scheme is now clear and useful for other non-specialists working in the area of natural protein evolution and epistasis (earlier we could not find studies of protein epistasis explaining these effects in a general and clear manner).

- I am confused by the abstract-level mention of the epistasis calculations. These seem quite routine.

Answer: We agree that calculating epistatic effects is trivial for double mutants, but we disagree that it is routine for more complex cases. For example, for a protein with 4 or more mutations that has been deconvoluted completely, the calculation effort can be very time-consuming, particularly when measuring more than one parameter or trait. In our study, we had a total of 8 mutants, each tested with 7 different parameters. Unfortunately, we were not able to find a public web server or computational program to calculate epistatic effects in an automated manner. For this reason, we created a program that could be used by anyone,

thus potentially making our study interesting beyond the community of directed evolution in general and of P450 in particular. Moreover, we have updated our server with additional examples (e.g., Ref. 67 and others) to enable a broader utilization of this tool. The additional examples have been added to the SI.

While utilizing our server to include new examples, we realized that our script had some issues, and that some of the calculations previously done for 4 parameters (TTN, TTF, PFR, NCR) were not appropriate because the formula used was wrong. We ran again the script with the right formula to obtain the correct values, which have been updated in Supplementary Tables 5-8. We also noticed that the values of the coupling efficiency parameter were calculated with those specific for 2 β -hydroxytestosterone instead of all products. For this reason, we re-calculated the epistatic effects for the coupling between NADPH and all products. To avoid misunderstandings in parameter definition, we also added in the title of Supplementary Tables 5-9 the word “total” or “2 β -hydroxytestosterone” when referring to all products or the main one, respectively.

While repeating all calculations of epistatic effects, we also realized that some data was missing in Supplementary Figure 1 to enable others to perform the calculations. For this reason, we added columns with the data for time, PFR, NCR and NADPH leak rate (NLR), which is the value needed to determine how much NADPH is wasted in the absence of substrate. The sum of the NLR and NCR (specific for product 2 β -hydroxytestosterone) results in total NADPH consumed.

Finally, the following changes were updated in the server and script (Github) and now they are fully functional:

- Added more explanations about the server to the user.
- Added more examples from other groups (e.g., evolutionary biologists) with complete deconvoluted datasets, but also added an example with partial deconvoluted variants.
- Corrected several lines of codes that were problematic.
- Added the option of choosing arithmetic mean or median depending on replicate number.
- Refactored the way graph results of epistatic effects were downloaded.

Reviewer #4:

Epistasis within a protein is supposed if the combined effect of several mutations deviates from the sum predicted by adding their individual effects. The aim of this study was to analyse and to quantify effects induced by all mutational pathways possible during protein engineering of P450 BM3 for the selective 15 β -hydroxylation of testosterone. To this end, eight possible mutants were produced, characterized regarding their activity and regioselectivity. A developed computational program has revealed pervasive positive epistatic effects. Quantum mechanics and MD simulations were applied to link protein epistasis and conformational changes in P450 BM3.

The manuscript provides new insights into the laboratory evolution of P450 BM3. The findings are of interest to the P450 community and the wider field.

Answer: We thank the reviewer for this positive statement.

Several questions remained open:

1. According to Fig. 2 and Supplementary Table 1, the mutants -II (T49I/Y51I/F87A) and III (R47I/T49I/Y51I/F87A) have very similar properties, except for initial rates and, correspondingly, TTF. C2-regioselectivity of these mutants is very high (85% and 91%, respectively). However, pose 15 was found highly unstable and the substrate rapidly rotates to C2 close to the Cpd I only in R47I/T49I/Y51I/F87A. Does it mean that the mutation R47I has such a strong effect on substrate flexibility in the binding pocket?

Answer: R47I mutation is placed on the β 1-2 strand and located in the pre-binding pocket (comprehended within the F helix/F-G loop, B' helix/B-B' loop, β 1 sheet, and A helices / a-A loop and the β 4 sheet), so it is a distal mutation with respect to the haem binding pocket. Therefore, R47I does not have a direct impact on the rotation of the substrate in III, which takes place in the haem pocket. Mutation R47I alters the enzyme conformational dynamics and induces long-range effects, which: (i) increase the haem pocket from 143 Å³ (-II apo-mutant, data not shown in the main text) to 235 Å³ (III apo-mutant) allowing substrate rotation and (ii) trigger substrate rotation through interactions with A87, T260, G265, T327, A330 (see description of Supplementary Video 1). To further investigate the specific effect of R47I mutation on substrate flexibility

inside the haem pocket, we have performed a PCA analysis on the substrate-bound MD trajectories of mutant -II and III finding that pc2 indeed describes an increased flexibility of residues A87, T260, G265 and T327 in mutant III (Figure R4b) as compared to -II (Figure R4a). We have included these new PCA analysis on the substrate-bound MD trajectories in the supporting information of the revised manuscript (Supplementary Figure 8), as well as added a short explanation of the effect of R47I mutation on substrate flexibility in the main text:

“Notwithstanding, mutant -II and III only differ for the R47I mutation in the latter case, yet the re-orientation of the substrate from pose 15 to pose 2 is observed only during the MD simulation of mutant III. To further investigate the specific effect of R47I mutation on substrate rotation inside the haem pocket, we performed a PC Analysis on the substrate-bound MD trajectories of mutant -II and III, finding that pc2 indeed describes an increased flexibility of residues A87, T260, G265 and T327 in mutant III, as compared to -II (Supplementary Fig. 8). Thus, R47I may modulate via long-range conformational dynamic effect the flexibility of such residues, which have been shown to be instrumental to promote substrate re-orientation in mutant III (see Supplementary Video 1)”

2. If the regioselectivity in all variants is determined by the orientation adopted by the substrate while accessing the active site, and only in the variant III the substrate rotates inside the active site bringing C2 close to the Cpd I, does it mean that the arginine at position 47 has a strong impact on the substrate orientation (or re-orientation) because it is the only difference between -II and III.

Answer: We thank referee 4 for his/her comments on the role played by R47I mutation. In the previous point, we addressed his/her concern about the effect of R47 and I47 on the rotation of the substrate when it is located in the binding pocket (haem active site). Regarding the regioselectivity being determined by the orientation of the substrate while accessing the active site (i.e. when it is still in the pre-binding pocket), we based our analysis on the aMD simulation of mutant I--, which describes the binding choreography of the substrate, entailing a stepwise mechanism of (i) recognition-fetching and (2) binding of testosterone. In the first step, the substrate is placed in the pre-binding pocket, continuously reorienting until the β_4 sheet retreats acting as a gate (second step) and allowing substrate progression toward the haem active site. We observed that the orientation of the testosterone while accessing the haem pocket (which consequently dictates its final binding pose) is determined by the hydrogen bond interaction between its carbonyl group and Y51, constraining the substrate in such a way that it can only progress into the active site pocket pointing its C15 toward O=Fe(haem) (lines 244-246 and Supplementary Figure 10).

The capability of both R47/Y51 to respectively form ion pair/h-bond interaction with long-chain fatty acids (e.g. palmitoleic acid) it is well-known since the first characterization of the substrate-bound P450-BM3 haem domain (Li and Poulos, *Nat. Struct. Biol.* 1997, 2, 140-146) and it was predicted even earlier by Ruettinger and Fulco (*JBC*, 1981, 11, 5728-5734). In particular, in this latter work, the authors found that when the fatty acid carboxylate is protonated at lower pH, the rate of epoxidation relative to hydroxylation increases, as the substrate-protein interaction involving R47 and Y51 is weakened. With the carboxylate group unrestricted, the fatty acid substrate can move further exposing its double bond to the O=Fe(haem) unit to undergo epoxidation instead of exposing its ω -carbons atom and undergoing hydroxylation. However, it should be noted that substrates like long-chain fatty acids are long enough to interact with both R47 and Y51 with their terminus-ends while they are bound in the active site, whereas shorter substrates like the one we used in our study (i.e. testosterone) may interact with these residues only when they are in the pre-binding pocket, and ultimately only with the closer Y51 when the β_4 sheet retreats and the substrate moves forward to the haem. Finally, Li and Poulos (*Nat. Struct. Biol.* 1997, 2, 140-146) observed that R47 is rather ill defined in the crystal structure, showing also a higher root mean square deviation (RMSD) than Y51 (Fig 3 in their paper), which is instead well ordered. They speculated that R47-substrate interaction may be flexible, transient and perhaps not very critical. Hence, based on these previous findings and our aMD simulation of mutant I--, we strongly believe that only Y51 has a strong impact on substrate orientation when accessing the haem pocket. Indeed, when this residue is mutated to an apolar Ile (in all the possible combination of mutation, i.e. -I, I-I, -II and III) the potential polar interaction with the carbonyl group of testosterone is deleted, thus changing the regioselectivity.

We have included a short description of the role of R47 and Y51 on substrate orientation in the binding pocket in the SI as well as added the following sentences in the main text and the Li/Poulos reference as 52:

“It should be noted that in previous studies, R47 and especially Y51 were found to interact with the terminus end of long-chain fatty acids while bound at the P450_{BM3} active site⁵². Such direct interaction with testosterone and Y51 is only possible at the pre-binding pocket, which is lost after the retreat of β 4 sheet allowing substrate access to the haem pocket.”

3. If no substrate rotation was observed during MD simulations of the C2-selective mutants, does it mean that starting pose 2 was used? Please comment on it. What happens if you start with pose 2 during MD simulations with the mutant III?

Answer: For each mutant, 3 independent replicas of MD simulation have been performed either starting from pose 2 (red color in the KDE plots, Fig. 3 and Supplementary Figures 6-7) or pose 15 (blue color in the same KDE plots). Substrate rotation was not observed in any of the MD simulations, with the exception of the 1st replica of mutant III starting from pose 15 (Figure 3). The MD simulations starting from pose 2 in mutant III lead to a stable substrate pose in all the three replicas. In addition to that, nearest attack conformations (considering the QM geometric parameters) were sampled for pose 2. In contrast, in the first MD replica (Figure 3) starting with pose 15, substrate rotation towards pose 2 was observed. In the remaining two replicas that start from pose 15 (Supplementary Figures 6,7), although substrate rotation was not observed, a reduced number of nearest attack conformation (considering the QM geometric parameters) was visited with respect to pose 2. We have added the following sentences in the main text to clarify this point:

“Given the identification of comparable reaction barriers for hydrogen-atom abstraction from C2 and C15 by using Density Functional Theory (DFT) calculations on truncated models (difference of $<1.0 \text{ kcal} \cdot \text{mol}^{-1}$, see Supplementary Note 2 and Supplementary Fig. 3), we carried out MD simulations starting, for each mutant, from pose 15 and from pose 2 (i.e. positioning C15 or C2 closer to the Fe=O, respectively) to analyse whether the binding pose of **1** in the active site determines the experimentally observed selectivity (Fig. 3 and Supplementary Note 3).”

“This is even more dramatic in variant III, in which pose 15 is highly unstable and **1** rapidly rotates to position C2 close to the catalytic Cpd I for 2 β -hydroxylation (Supplementary Video 1). Instead, pose 2 in variant III is stable and adopts NAC conformations in all MD replicas (Fig. 3d and Supplementary Figs. 6 and 7).”

4. According to aMD simulations, enzyme regioselectivity is determined by the orientation of the substrate when accessing the catalytic site during the second step of the binding pathway. How does it relate to the observed rotation of the substrate in the variant III? Please comment on it.

Answer: As stated in the manuscript (lines 240-241): “Our aMD simulations show that the orientation of the substrate when accessing the catalytic site during the second step of the binding pathway dictates selectivity.” Indeed, during the binding event of the substrate (aMD simulation of mutant I--) we observed a stepwise binding pathway. In the first step the substrate is fetched by the enzyme, forming a long living intermediate (similarly to what is observed by Mondal et al. for P450cam, *JACS* 2018, 140, 50, 17743–17752) in the “pre-binding” pocket located within the F helix/F-G loop, B' helix/B-B' loop, β 1 sheet, and A helices / a-A loop and the β 4 sheet. In this first step we observed the substrate continuously reorienting, until the β 4 sheet retreats (second step of the binding) and allows the substrate to progress to the haem pocket (located within the B', F, I helices, β 4 sheet, and the B-C loop). Once inside the haem pocket, substrate rotation was not observed, thus indicating that the orientation assumed by the substrate when entering the haem pocket will be its final binding pose ultimately dictating selectivity. It should be noted that C15 and C2 of testosterone are located at the two ends of the molecule, so that switching from pose 15 to pose 2 implies a complete 180° rotation of the molecule. This rotation was neither observed in the aMD binding trajectory of I--, nor in MD simulation of any other mutant, except in one replica of mutant III, in which starting from pose 15, we observed the substrate to rotate to a stable pose 2. Indeed, our volume calculations suggest that only mutant III possesses a bigger haem pocket which may allow the reorientation of the substrate, even during the second step of the substrate binding mechanism. Within this picture, we can conclude that the orientation of the substrate (entering with its C15 or its C2) at the moment in which the β 4 sheet retreats determines its final pose in the haem pocket and thus the C2 or C15 hydroxylation. However, this is not crucial for mutant III, since in this particular case the reorientation from pose 15 to pose 2 in the second step of the binding pathway is possible thanks to the bigger volume of the haem pocket.

It should be noted also that in mutant III the “opposite” rotation from pose 2 to pose 15 was never observed during MD simulation, but this possibility could not be ruled out.

We have revised the main text to better clarify these findings:

“We hypothesised that in all variants, **except III**, selectivity must be determined by the orientation adopted by the substrate while accessing the haem cavity.”

“Our aMD simulations **indicate** that the orientation of the substrate when accessing the catalytic site during the second step of the binding pathway dictates selectivity. **Once inside the haem pocket, substrate rotation was not observed in mutant I--**, thus predicting that the orientation assumed by the substrate when entering the haem site ultimately governs selectivity. In fact, the previous substrate-bound MD simulations suggest that only variant III has a sufficiently wide active site pocket for allowing substrate rotation. These findings indicate that in all variants, **except III**, selectivity is determined by the orientation adopted by the substrate while entering the haem pocket.”

We hope that these changes will make our paper suitable for publication in *Nature Communications*.

Yours sincerely,

Sílvia Osuna
Institut de Química Computacional i Catàlisi
Universitat de Girona,
Carrer Maria Aurèlia Capmany 69
17003 Girona, Spain

Manfred T. Reetz
Max-Planck-Institut für Kohlenforschung
Kaiser-Wilhelm-Platz 1,
45470 Mülheim an der Ruhr
Germany

REVIEWERS' COMMENTS

Reviewer #1 (Remarks to the Author):

As I outlined in my initial review, I am overall very positively inclined towards this work, which I think makes an important and valuable contribution to the field. I had a number of issues I felt needed addressing however, which the authors have done to my satisfaction. Therefore I am happy with this work and believe it will be of broad interest and high value to both experts in the field and generalists interested in protein design and enzyme evolution, as well as trying to understand enzyme function more generally.

Reviewer #2 (Remarks to the Author):

I am satisfied with the thorough revisions provided by the authors. I believe this manuscript will be of great interest to the community and recommend its publication.

Reviewer #4 (Remarks to the Author):

In the revised version of the manuscript the authors have addressed all my concerns. Furthermore, more detailed and convincing explanations have been included and additional data provided. The findings are of interest not only to the P450 community but to the wider field. I recommend to accept the manuscript for publication.